# In Vivo Neuroprotective Effects of Alpinetin Against Experimental Ischemic Stroke Damage Through Antioxidant and Anti-Inflammatory Mechanisms

**DOI:** 10.3390/ijms26115093

**Published:** 2025-05-26

**Authors:** Ratchaniporn Kongsui, Sitthisak Thongrong, Jinatta Jittiwat

**Affiliations:** 1Division of Physiology, School of Medical Sciences, University of Phayao, Phayao 56000, Thailand; ratchaniporn.ko@up.ac.th; 2Division of Anatomy, School of Medical Sciences, University of Phayao, Phayao 56000, Thailand; sitthisak.th@up.ac.th; 3Faculty of Medicine, Mahasarakham University, Mahasarakham 44000, Thailand

**Keywords:** ischemic stroke, middle cerebral artery, alpinetin, antioxidant, anti-inflammatory

## Abstract

Ischemic stroke is the most common type of stroke and poses a major global health challenge due to its high mortality and lasting disability impact. The onset and progression of ischemic stroke are largely linked to oxidative stress and inflammatory responses. Alpinetin, a natural flavonoid found in the ginger family, exhibits various pharmacological properties, including antioxidant and anti-inflammatory activities. In this study, the neuroprotective potential of alpinetin in attenuating oxidative stress and inflammation against cerebral ischemic stroke was evaluated. Ninety male Wistar rats were randomly assigned to the sham operation group, the Rt.MCAO group, the Rt.MCAO+piracetam group, and the Rt.MCAO+alpinetin groups (25, 50, and 100 mg/kg BW). Cerebral infarction size, neuronal density, and antioxidant and anti-inflammatory activities were measured. Three days of treatment with alpinetin markedly reduced the infarct volume by 30% compared to the Rt.MCAO+vehicle-treated group. Additionally, rats treated with alpinetin exhibited a significant increase in neuronal density in the cortex, as well as in the CA1 and CA3 regions of the hippocampus. Furthermore, treatment with alpinetin ameliorated both the Rt.MCAO-induced increase in malondialdehyde (MDA) activity and the Rt.MCAO-induced decrease in catalase (CAT), glutathione peroxidase (GSH-Px), and superoxide dismutase (SOD) activities in the cortex and hippocampus. Moreover, COX-2 and IL-6 protein levels were assessed using western blotting. The results showed that treatment with alpinetin (100 mg/kg BW) significantly reduced the expression levels of COX-2 and IL-6 in both the cortex and hippocampus. Our findings suggest that alpinetin significantly mitigates the effects of cerebral ischemia-induced brain damage through its antioxidant and anti-inflammatory properties and could potentially be developed as a therapeutic agent for stroke treatment.

## 1. Introduction

Ischemic stroke is the most frequently occurring form of stroke, accounting for around 87% of all cases [1]. It represents a serious global health concern due to its substantial contribution to death rates and long-term disability [2]. Ischemic stroke occurs when a blockage in a cerebral artery disrupts blood flow, depriving brain tissue of essential oxygen and nutrients. This acute event triggers a cascade of cellular and molecular disturbances, including excitotoxicity, oxidative stress, and inflammation, which can lead to irreversible neuronal damage. Understanding these pathological mechanisms is crucial for developing targeted therapeutic strategies to mitigate the detrimental effects of stroke and improve clinical outcomes. Moreover, this energy deficiency initiates a series of harmful biochemical and molecular changes. One of the early responses is a shift to anaerobic metabolism, which results in the excessive production of reactive oxygen species (ROS). These molecules contribute to oxidative stress by damaging essential cellular components such as lipids, proteins, and DNA, leading to increased neuronal injury and cell death [3,4]. The oxidative imbalance can also weaken the blood–brain barrier (BBB), trigger inflammatory processes, and extend damage to surrounding brain areas [5,6]. Antioxidant enzymes such as catalase (CAT), glutathione peroxidase (GSH-Px), and superoxide dismutase (SOD) play a critical role in neutralizing ROS and protecting neurons from oxidative damage [7,8,9]. In addition to oxidative stress, inflammation following ischemia involves the activation of microglia and astrocytes, which release pro-inflammatory cytokines including interleukin-6 (IL-6) [10,11]. Cyclooxygenase-2 (COX-2) also becomes upregulated and contributes to inflammation through the production of prostaglandins, which worsen neural damage [12]. Increased levels of IL-6 and COX-2 promote immune cell infiltration into the brain, maintain the inflammatory state, and delay tissue recovery [10,13]. Therefore, targeting these molecules may offer a therapeutic approach to reduce secondary injury and improve recovery after stroke [10,13,14,15].

Flavonoids, a class of polyphenolic compounds widely found in fruits, vegetables, and medicinal herbs, have demonstrated notable neuroprotective effects [16]. Their actions include neutralizing reactive oxygen species (ROS), modulating cellular signaling pathways, reducing inflammatory responses, exhibiting anticancer activity, and mitigating excitotoxic damage [17,18,19,20,21]. Experimental studies have shown that compounds such as prunin, quercetin, and baicalein can reduce brain infarct size, improve behavioral outcomes, and enhance antioxidant enzyme activity in ischemic stroke models [22,23,24]. These findings suggest that flavonoids hold promise as therapeutic agents for neuroprotection. Following intraperitoneal (i.p.) administration of alpinetin at a dose of 50 mg/kg, its elimination half-life (T½) was found to be approximately 9 h. This route of administration appears to enhance systemic exposure and may support improved delivery to the brain [25]. Alpinetin, a flavonoid found in the ginger family, exhibits various pharmacological properties, including antioxidant [26], anti-inflammatory [26,27,28], anti-cancer [29], liver-protective [20], and lipid metabolism-regulating effects [25]. Previous studies found that alpinetin notably lowered markers of oxidative and nitrosative stress in the liver, including malondialdehyde (MDA) and other reactive oxygen species [26]. It also boosted antioxidant defenses by increasing levels of SOD and glutathione, thereby strengthening the liver’s ability to combat oxidative damage [26]. Moreover, this compound may inhibit inflammation by downregulating pro-inflammatory cytokines such as tumor necrosis factor-α (TNF-α), IL-1β, IL-4, and IL-6, while also reducing the activation of immune cells involved in the inflammatory response [26,28]. Oxidative stress emerges early after ischemic stroke and intensifies over the first 72 h [4]. During this phase, activated microglia and infiltrating immune cells produce large amounts of ROS via enzymes like myeloperoxidase and inducible nitric oxide synthase [4,30]. This ROS surge exacerbates damage in the penumbra, contributing to infarct expansion and hindering recovery [4]. Targeting oxidative stress in this window is crucial for reducing neuronal injury and improving outcomes. Considering the significant involvement of oxidative stress and inflammation in cerebral ischemia within 72 h, along with the reported therapeutic properties of alpinetin, this study aimed to determine the effect of alpinetin on brain infarct volume, neuronal loss, and antioxidant and anti-inflammatory activities using an animal model of focal cerebral ischemia.

## 2. Results

### 2.1. Alpinetin Alleviated Ischemic Stroke-Induced Cerebral Infarct Volume

To investigate how alpinetin affects ischemic stroke-induced brain damage, infarct volume in the brain was investigated. All experimental animals were subjected to blockage of the right middle cerebral artery (Rt.MCAO) by inserting a 4.0 nylon filament approximately 17 mm through the right common carotid artery. Treatments included intraperitoneal (i.p.) administration of either the vehicle, piracetam (250 mg/kg BW), or different doses of alpinetin (25, 50, or 100 mg/kg BW) once daily for 3 consecutive days. Following the treatment period, brain infarct volume was assessed using 2,3,5-triphenyltetrazolium chloride (TTC) staining. As shown in Figure 1, intraperitoneal administration of piracetam and alpinetin for 3 days following occlusion significantly reduced infarct volume compared to the Rt.MCAO+vehicle group (*p* < 0.05). The sham-operated group showed no detectable infarct.

### 2.2. Alpinetin Improved Ischemic Stroke-Induced Neuronal Loss in the Cortex and Hippocampus

To further determine the neuroprotective potential of alpinetin on neural function, the neuronal cell density in the cortex and the CA1 and CA3 regions of the hippocampus was evaluated. The results show that rats treated with piracetam at 250 mg/kg BW or alpinetin (at 50 or 100 mg/kg BW) for 3 consecutive days exhibited notably increased neuronal cell densities in the cortex and the CA1 and CA3 hippocampal regions compared to the Rt.MCAO+vehicle group (*p* < 0.05). Additionally, rats treated with alpinetin at a dose of 25 mg/kg BW for 3 days showed a significant increase in neuronal cell density in the CA1 and CA3 regions of the hippocampus, but not in the cortex, compared to the Rt.MCAO+vehicle group (Figure 2).

### 2.3. Alpinetin Ameliorated Ischemic Stroke-Induced Excessive Production of Malondialdehyde (MDA) in the Cortex and Hippocampus

To establish if alpinetin mitigates brain damage by reducing oxidative stress, MDA levels were measured in the cortex and hippocampus of rats subjected to Rt.MCAO. The animals were treated i.p. once daily for 3 consecutive days with either vehicle, piracetam (250 mg/kg BW), or alpinetin (25, 50, or 100 mg/kg BW). The results revealed that treatment with piracetam at 250 mg/kg BW or alpinetin at 50 or 100 mg/kg BW significantly reduced MDA levels in both the cortex and hippocampus compared to the Rt.MCAO+vehicle group (*p* < 0.05). In contrast, alpinetin at 25 mg/kg BW resulted in a statistically significant reduction in MDA levels only in the hippocampus and not in the cortex (Figure 3).

### 2.4. Alpinetin Increased Catalase (CAT) Activity in the Cortex and Hippocampus

To determine if the neuroprotective effects of alpinetin in Rt.MCAO rats were mediated by its antioxidant effects, CAT levels in the cortex and hippocampus were measured. Treatment with piracetam at 250 mg/kg BW or alpinetin at 100 mg/kg BW were found to significantly increase CAT levels in both the cortex and hippocampus compared to the vehicle-treated group (Rt.MCAO+vehicle) (*p* < 0.05 for all comparisons; Figure 4). In contrast, there was no significant difference in CAT level in either brain region in rats treated with 25 or 50 mg/kg BW of alpinetin compared to the Rt.MCAO+vehicle group.

### 2.5. Alpinetin Enhanced Glutathione Peroxidase (GSH-Px) Levels in the Cortex and Hippocampus

Following the completion of the experiment, we further investigated if alpinetin modulated changes in cortical and hippocampal GSH-Px levels in rats. Rats treated with piracetam at 250 mg/kg BW or alpinetin at 50 or 100 mg/kg BW were found to exhibit a statistically significant increase in GSH-Px levels in both the cortex and hippocampus compared to the Rt.MCAO+vehicle group (*p* < 0.05 for all comparisons; Figure 5). In contrast, rats receiving alpinetin at a dose of 25 mg/kg BW showed a significant increase in GSH-Px levels in the hippocampus, but not in the cortex, compared to the vehicle-treated group (Rt.MCAO+vehicle) (*p* < 0.05).

### 2.6. Alpinetin Improved Superoxide Dismutase (SOD) Levels in the Cortex and Hippocampus

To confirm the antioxidant effect of alpinetin, the cortex and hippocampus regions of rat brains were analyzed for SOD levels. The results indicated that administration of piracetam at 250 mg/kg BW or alpinetin at 100 mg/kg BW significantly increased SOD levels in both the cortex and hippocampus compared to the vehicle-treated group (Rt.MCAO+vehicle) (*p* < 0.05 for all comparisons; Figure 6). Conversely, treatment with alpinetin at doses of 25 and 50 mg/kg BW resulted in no notable changes in either the cortex or hippocampus compared to the Rt.MCAO+vehicle group.

### 2.7. Alpinetin Regulated the Expression of Cyclooxygenase-2 (COX-2) and Interleukin-6 (IL-6) in the Cortex and Hippocampus

In the next set of experiments, the mechanism by which alpinetin modulates COX-2 and IL-6 expression (anti-inflammatory pathway) was investigated, and the results are shown in Figure 7. Rats treated with alpinetin at a dose of 100 mg/kg BW were selected for further study, as this dose demonstrated the most effective neuroprotective effects against ischemic brain injury in this study. The cortical and hippocampal regions of the brain were analyzed by Western blot technique. Brain tissues were homogenized using N-PER^TM^ extraction reagent (Thermo Fisher Scientific, Inc., cat. no. 87792, Waltham, MA USA), and COX-2 and IL-6 expression levels were assessed. The results showed that treatment with piracetam (250 mg/kg BW) or alpinetin (100 mg/kg BW) for 3 consecutive days significantly reduced COX-2 and IL-6 expression levels in both the cortex and hippocampus compared to the Rt.MCAO+vehicle (*p* < 0.05).

## 3. Discussion

Under physiological conditions, the brain receives a blood flow of about 55 mL per 100 g of tissue each minute. During an episode of cerebral ischemia, however, this flow can decline drastically, falling below 10 mL/100 g/min [31]. The most common cause of stroke, responsible for roughly 87% of cases, is the narrowing or obstruction of cerebral arteries, often due to thrombi that originate elsewhere and subsequently become trapped within brain vessels [1]. Major risk factors for this condition include hypertension, diabetes mellitus, elevated blood lipids, and smoking [32]. Within the initial 72 h following a stroke, a complex series of pathological processes unfolds, culminating in neuronal injury and inflammatory reactions. The interruption of cerebral blood flow leads to ischemia-induced neuronal death, which then triggers a chain of biochemical events. Among these is the excessive release of excitatory neurotransmitters, particularly glutamate, which exacerbates neuronal damage by promoting calcium influx, generating oxidative stress, and initiating inflammatory pathways [33,34,35]. Moreover, following an ischemic stroke, oxidative stress rapidly becomes a key contributor to brain injury and progressively intensifies over the ensuing days. During the secondary phase of neuroinflammation—occurring after the initial ischemic event—there is a significant rise in the production of ROS. Between roughly 18 and 72 h post-stroke onset, activated microglia and infiltrating immune cells, such as macrophages and neutrophils, generate ROS through enzymatic systems like myeloperoxidase and inducible nitric oxide synthase [4,30]. This surge, often referred to as a “respiratory burst”, exacerbates neuronal damage, particularly within the penumbral region adjacent to the ischemic core, thereby worsening cellular injury and expanding the infarct size [36]. Considering the significant involvement of oxidative stress and inflammation in cerebral ischemia within 72 h, along with the reported therapeutic properties of alpinetin, this study aimed to determine the effect of alpinetin on brain infarct volume, neuronal loss, and antioxidant and anti-inflammatory activities using an animal model of focal cerebral ischemia induced by permanent blockage of the Rt. MCAO.

This study investigated the effects of alpinetin on infarct volume in 8-week-old male rats weighing 250–300 g. All experimental animals underwent Rt.MCAO via surgical insertion of a 4.0 nylon filament approximately 17 mm into the right common carotid artery. The rats were treated with either a vehicle (DMSO), piracetam at a dose of 250 mg/kg BW, or alpinetin at doses of 25, 50, or 100 mg/kg BW. All treatments were administered i.p. once daily for three consecutive days. After the experimental period, infarct volume was assessed using TTC staining. The results revealed that both piracetam and alpinetin significantly reduced infarct volume compared to the Rt.MCAO + vehicle group (*p* < 0.05). Additionally, this study examined the effects of alpinetin on neuronal density in the cortex and the hippocampal CA1 and CA3 regions in rats subjected to Rt.MCAO. It was found that rats treated with piracetam at 250 mg/kg BW or alpinetin at 50 or 100 mg/kg BW for 3 days exhibited significantly higher neuronal densities in the cortex and hippocampal CA1 and CA3 areas compared to the vehicle-treated group (*p* < 0.05, all compared with Rt.MCAO+vehicle). Moreover, treatment with alpinetin at 25 mg/kg BW for 3 days significantly increased neuronal density only in the hippocampal CA1 and CA3 regions (*p* < 0.05, compared with Rt.MCAO+vehicle). Alpinetin has been shown to exert notable vasodilatory effects, supporting its potential as a therapeutic agent in cardiovascular disorders [37]. Research indicates that it facilitates vascular smooth muscle relaxation through both endothelium-dependent and -independent pathways [29,38]. One of its key actions involves enhancing nitric oxide (NO) production, a vital mediator of vasodilation [39]. Simultaneously, alpinetin suppresses calcium entry and reduces intracellular calcium mobilization within vascular smooth muscle cells. It also interferes with excitation–contraction coupling by modulating protein kinase C activity, further contributing to its vasodilatory function [38]. Through these mechanisms, alpinetin lowers vascular resistance and enhances circulatory efficiency, suggesting potential benefits for stroke conditions. In our study, the 25 mg/kg dose of alpinetin exhibited region-specific differences, showing limited efficacy in the hippocampus compared to the cortex. This variability may stem from differential regional vulnerability to ischemic injury, blood–brain barrier permeability, or metabolic processing of alpinetin. Such region- and dose-specific responses have also been reported with other flavonoids. For instance, quercetin has been shown to exert stronger neuroprotective effects in the cortex than in hippocampal regions at lower doses, likely due to differences in bioavailability and oxidative stress profiles across regions [40]. Similarly, resveratrol has demonstrated variable protective outcomes depending on dose, timing, and brain region targeted [41,42]. These parallels support the idea that alpinetin’s effects may be consistent with broader trends observed among flavonoids, and further studies will help determine the optimal dosing for region-specific protection.

During ischemic stroke, there is a substantial increase in the production of free radicals, which negatively affects the body’s endogenous antioxidant defense system. This results in an imbalance between free radicals and antioxidants. The excessive accumulation of free radicals causes cellular damage and leads to neuronal cell death [39]. However, in the surrounding area known as the penumbra—where partial blood flow is still present—neuronal cells are not immediately destroyed but are still affected by increased oxidative stress. This is primarily due to the reduced activity of key antioxidant enzymes, such as glutathione peroxidase, catalase, and superoxide dismutase [2,8,14,43]. The elevated levels of free radicals in this region further contribute to neuronal death, ultimately impairing the function of brain areas affected by cell loss. MDA is a byproduct of lipid peroxidation and serves as a biomarker of oxidative stress. Elevated MDA levels indicate oxidative damage to lipids within cells and tissues. Following a stroke, MDA levels may increase due to a cascade of events that lead to oxidative stress and lipid oxidation in the brain. This study aimed to investigate the effects of alpinetin on changes in MDA levels in male rats (8 weeks old, weighing 250–300 g) subjected to Rt.MCAO. The animals were treated with either the vehicle, piracetam at a dose of 250 mg/kg BW, or alpinetin at doses of 25, 50, or 100 mg/kg BW via i.p. injection once daily for three consecutive days. After the treatment period, brain tissues were collected to assess changes in MDA levels. The results showed that treatment with piracetam (250 mg/kg BW) or alpinetin at doses of 50 and 100 mg/kg BW significantly reduced MDA levels in both the cortex and hippocampus regions. In contrast, treatment with alpinetin at 25 mg/kg BW resulted in a statistically significant reduction of MDA levels only in the hippocampus compared to the Rt.MCAO+vehicle group (*p* < 0.05). In addition, this study also aimed to investigate the effects of alpinetin on the activity of antioxidant enzymes, including CAT, GSH-Px, and SOD, in rats subjected to Rt.MCAO. The results revealed that treatment with piracetam at a dose of 250 mg/kg BW or alpinetin at doses of 50 or 100 mg/kg BW significantly increased the levels of GSH-Px and CAT in both the cortex and hippocampus compared to the Rt.MCAO+vehicle group (*p* < 0.05). Regarding changes in SOD levels in the cortex and hippocampus, it was found that treatment with piracetam at a dose of 250 mg/kg BW or alpinetin at doses of 50 or 100 mg/kg BW significantly increased SOD levels in both brain regions when compared to the Rt.MCAO+vehicle group (*p* < 0.05). These findings are consistent with a previous study showing that alpinetin has antioxidant properties, including reducing levels of MDA and enhancing the activity of antioxidant enzymes such as SOD, GSH-Px, and CAT in mice with experimentally induced liver fibrosis [26]. Moreover, alpinetin exhibits strong antioxidant activity, largely due to its flavonoid-based structure, which enables it to donate phenolic hydrogen atoms and scavenge reactive free radicals effectively. Additionally, alpinetin has been shown to upregulate superoxide dismutase 1 (SOD1), heme oxygenase-1 (HO-1), and the activity of the transcription factor nuclear factor erythroid 2–related factor 2 (Nrf2) [28], which contributes to reducing disease severity in a metabolic dysfunction-associated fatty liver disease (MASLD) [44]. The Nrf2 pathway is essential in protecting cells against oxidative stress by regulating the expression of antioxidant enzymes, including SOD, CAT, and GSH-Px [45]. Under oxidative stress conditions, such as those induced by Rt.MCAO, the Nrf2–Keap1 interaction is disrupted, allowing Nrf2 to translocate into the nucleus and activate the transcription of antioxidant genes via the antioxidant response element (ARE) [46]. In this study, Rt.MCAO led to a marked reduction in hippocampal SOD, CAT, and GSH-Px levels, reflecting a compromised antioxidant defense system, which aligns with previous findings highlighting the overproduction of ROS during cerebral ischemia [47,48]. Treatment with alpinetin and piracetam effectively reversed these reductions, indicating that both agents may enhance the antioxidant response by modulating Nrf2 activity. Alpinetin, a naturally occurring flavonoid, has been reported to facilitate Nrf2 nuclear translocation and stimulate antioxidant gene expression [49]. Meanwhile, piracetam—primarily known for its neuroprotective effects— may contribute to Nrf2 regulation indirectly by stabilizing mitochondrial function and reducing oxidative burden [50]. Altogether, these findings suggest that the neuroprotective effects of alpinetin and piracetam may be mediated, at least in part, through the enhancement of Nrf2-driven antioxidant defenses. The region-specific effect of alpinetin on MDA levels may be attributed to differences in baseline oxidative stress, metabolic activity, or antioxidant defense mechanisms between the hippocampus and cortex. The hippocampus is particularly susceptible to oxidative damage during ischemia, which may make it more responsive to antioxidant intervention at lower doses. Furthermore, the activation of SOD may require a higher threshold of alpinetin concentration due to variations in enzyme regulation, cellular localization, or sensitivity to oxidative stress compared to GSH-Px and CAT. This observation may also reflect differences in the affinity of alpinetin or its metabolites for the molecular pathways regulating each enzyme. Further investigation is needed to clarify the underlying mechanisms of this differential response.

Piracetam, a well-known nootropic agent, has been reported to enhance cerebral blood flow in both infarcted regions and surrounding penumbral areas. While its neuroprotective mechanism(s) are yet to be definitively elucidated, it is hypothesized that piracetam interacts with the polar head groups of phospholipids in the cell membrane, thereby improving membrane fluidity. This enhancement may support essential membrane-associated functions, such as ATP synthesis, ultimately contributing to the preservation of neuronal viability [51]. Earlier research has indicated that administering piracetam intraperitoneally at doses of 250 or 500 mg/kg body weight at 6, 9, and 22 h following the onset of cerebral artery occlusion can significantly reduce infarct volume in ischemic brain tissue [52]. These results align with observations from our prior investigations [15,39]. Additionally, our previous reports have shown that piracetam provides protection against oxidative stress by enhancing the activity of antioxidant enzymes [43,53]. Due to its demonstrated ability to reduce brain infarct volume and enhance the activity of antioxidant enzymes following cerebral artery occlusion, piracetam was selected as a positive control in this study.

Previous studies have shown that anti-inflammatory cytokines, such as interleukin-10 (IL-10), may help protect against ischemia-induced neuronal death by inhibiting the action of pro-inflammatory cytokines, particularly tumor necrosis factor-α (TNF-α), thereby reducing the volume of brain infarction [54]. Additionally, the nuclear factor-kappa B (NF-κB) signaling pathway is a key mediator of the inflammatory cascade triggered by cerebral ischemia. During ischemic stroke, NF-κB becomes rapidly activated in various cell types, including neurons, glial cells, and endothelial cells. This activation promotes the expression of pro-inflammatory mediators such as cytokines, chemokines, adhesion molecules, and inflammatory enzymes like iNOS and COX-2 [55]. These changes contribute to the breakdown of the blood–brain barrier, infiltration of immune cells, and subsequent neuronal damage, all of which worsen ischemic outcomes [56]. Experimental studies have demonstrated that suppressing NF-κB activity can reduce infarct size and enhance neurological recovery, underscoring its therapeutic potential [57,58]. Therefore, targeting the NF-κB pathway represents a promising approach for limiting neuroinflammation and preserving brain tissue following stroke. Moreover, levels of pro-inflammatory cytokines such as TNF-α, interleukin-1 (IL-1), and interleukin-6 (IL-6) detected in the blood have been correlated with motor function and infarct volume [59]. This study investigated the effect of alpinetin on the expression of COX-2 and IL-6 (as part of the anti-inflammatory pathway) in the cortex and hippocampus using western blot analysis. The results showed that rats treated with alpinetin at a dose of 100 mg/kg BW once daily for three consecutive days following right middle cerebral artery occlusion (Rt.MCAO) exhibited a statistically significant reduction in COX-2 and IL-6 expression compared to the Rt.MCAO+vehicle group (*p* < 0.05). Notably, alpinetin has been shown to suppress neuroinflammation by modulating microglial activation and downregulating the JAK2/STAT3 signaling pathway [60]. It can also counteract microglia-induced production of ROS and the reduction of mitochondrial membrane potential (MMP) in PC12 neuronal cells. Moreover, an in vivo study indicates that alpinetin effectively reduces inflammation and neuronal cell death, while promoting axonal repair and enhancing motor function recovery [60].

Flavonoids are secondary metabolites belonging to the polyphenol group, commonly found in vegetables, fruits, and certain beverages. Their primary pharmacological property is antioxidant activity, as they help reduce oxidative stress [19,61]. Previous research has shown that neurodegenerative disorders, including stroke, are linked to increased free radical production and neuroinflammation. [62,63]. These elevated free radicals can stimulate the release of pro-inflammatory cytokines, leading to neuroinflammation and subsequent neuronal cell death [43,62,63]. Therefore, flavonoids help mitigate these pathological processes by reducing the formation of ROS. Alpinetin is a flavonoid compound that possesses a wide range of pharmacological properties, including antioxidant [26], anti-inflammatory [26,27,28], and anti-apoptotic effects [60]. It also offers protection to vascular smooth muscle cells (VSMCs) [64], exhibits anti-cancer activity by inhibiting tumor angiogenesis [29], and helps prevent chronic obstructive pulmonary disease (COPD) [65]. Additionally, alpinetin has protective effects on chondrocytes in osteoarthritis [27] and demonstrates anti-inflammatory activity in intestinal inflammation [66]. A study by He et al. (2016) investigated the effects of alpinetin on acute colitis in mice induced by dextran sulfate sodium (DSS) [67]. The study utilized alpinetin at doses of 25, 50, and 100 mg/kg BW and found that alpinetin could reduce inflammation by inhibiting the TLR4 and NLRP3 signaling pathways [68]. Moreover, Wu et al. (2020) investigated the anti-inflammatory potential of alpinetin in a mouse model of asthma. Their findings indicated that alpinetin, administered at doses of 25, 50, and 100 mg/kg body weight (BW), produced notable anti-inflammatory effects, with the highest dose (100 mg/kg) showing the greatest efficacy [28]. This is consistent with our current findings.

One limitation of this study is the short duration of outcome evaluation, which was limited to only 3 days post-treatment and may not fully capture the long-term neuroprotective effects and functional recovery following alpinetin administration in the Rt.MCAO model. Additionally, the exclusive use of male rats may restrict the applicability of the findings, as sex-related differences in stroke pathology and treatment response are increasingly recognized in preclinical research. Furthermore, this study did not include behavioral assessments to evaluate motor, sensory, or cognitive deficits following stroke. The absence of such functional outcome measures limits the ability to fully characterize the extent of neurological damage and recovery over time. Future research should extend follow-up periods, include both sexes, and incorporate comprehensive behavioral evaluations to enhance translational relevance and improve understanding of alpinetin’s therapeutic potential.

## 4. Materials and Methods

### 4.1. Treatment Substances

Alpinetin (PubChem ID: 154279), with a confirmed purity of 98% determined via high-performance liquid chromatography (HPLC), was obtained from Chengdu Biopurify Phytochemicals Ltd. (Sichuan, China) (Appendix A). The chemical structure of alpinetin (C_16_H_14_O_4_) is presented in Figure 8. Piracetam, used as a positive control in this study, was supplied by GlaxoSmithKline (Bangkok, Thailand) Ltd. Dimethyl sulfoxide (DMSO), used as the vehicle, was sourced from Thermo Fisher Scientific (catalog number: D/4121/PB15).

### 4.2. Animals

Male Wistar rats, aged eight weeks and weighing 250–300 g, served as the experimental animals. These animals were sourced from the Northeastern Laboratory Animal Center at Khon Kaen University (Khon Kaen, Thailand). Upon arrival, all rats were acclimatized for one week and housed in groups of five in standard metal cages measuring 37.5 × 48 × 21 cm^3^. They were maintained under controlled laboratory conditions, which included a consistent 12-h light–dark cycle, a relative humidity range of approximately 30–60%, and a stable temperature of 23 ± 2 °C. Throughout the study, the rats had unrestricted access to both drinking water and commercially available pellet food, available continuously. All protocols and procedures involving the use of these animals were reviewed and approved by the Institutional Animal Care and Use Committee at Khon Kaen University, Thailand (approval number: IACUC-KKU-95/64).

### 4.3. Rt.MACO Model

Prior to surgery, all rats underwent overnight fasting with unrestricted access to water. Anesthesia was initiated using 5% isoflurane and sustained with 1–3% isoflurane in 100% oxygen. Permanent focal cerebral ischemia was produced by occluding the right middle cerebral artery using an intraluminal method in accordance with standard procedures. A silicone-coated 4-0 monofilament (USS DGTM, United States Surgical; Tyco Healthcare Group LP, Norwalk, CT, USA) was inserted into the internal carotid artery to a depth of approximately 17 mm or until slight resistance was felt [64]. Following the procedure, the surgical incision was closed with sutures and disinfected using 10% povidone–iodine. In the sham group, the arteries were exposed in the same manner, but no filament was inserted. Humane endpoints included signs such as immobility, surgical wound infection, weight loss exceeding 20%, dehydration, respiratory distress, ongoing pain, lack of responsiveness to external stimuli, or bleeding from any orifice. No animals met humane endpoint criteria, and all survived through the 4-day observation period.

### 4.4. Animal Treatment

Ninety healthy male Wistar rats, as mentioned in the section above, were randomly divided into six experimental groups, with fifteen rats assigned to each group (n = 15), as follows (Table 1):Sham operation group: This group underwent sham surgery without insertion of the nylon filament and did not receive any treatment.Rt.MCAO+vehicle-treated group: In this group, all animals were subjected to Rt.MCAO by using the same procedure as described above. They received only the vehicle, which in this study was dimethyl sulfoxide (DMSO), administered i.p. at a volume of 0.5 mL once daily for three consecutive days. This group served as the vehicle control to evaluate the effects of the solvent.Rt.MCAO+piracetam 250 mg/kg BW-treated group: All animals in this group were subjected to Rt.MCAO and treated with piracetam at a dose of 250 mg/kg BW, administered i.p. once daily for three consecutive days. This group served as the positive control.Rt.MCAO+ALP 25 mg/kg BW-treated group: All animals in this group underwent Rt.MCAO and were treated with alpinetin at a dose of 25 mg/kg BW. The treatment was administered i.p. once daily for three consecutive days.Rt.MCAO+ALP 50 mg/kg BW-treated group: All animals in this group underwent Rt.MCAO and were treated with alpinetin at a dose of 50 mg/kg BW. The treatment was administered i.p. once daily for three consecutive days.Rt.MCAO+ALP 100 mg/kg BW-treated group: All animals in this group underwent Rt.MCAO and were treated with alpinetin at a dose of 100 mg/kg BW. The treatment was administered i.p. once daily for three consecutive days.

After completing the 3-day experimental period, infarct volumes were evaluated in 5 rats per group using TTC staining. Another set of 5 rats per group was used to assess neuronal density through cresyl violet staining of the cerebral cortex and hippocampus. The remaining 5 rats in each group underwent biochemical assays to measure MDA levels, CAT, GSH-Px and SOD activities in the brain cortex and hippocampus. Additionally, IL-6 and COX-2 expression levels were quantified in the cortex and hippocampus of rats receiving the most effective doses of alpinetin, which showed the greatest impact on infarct volume, neuronal density, and oxidative stress indicators.

### 4.5. Assessment of Brain Infarct Volume

At the end of the study, rats were perfused with cold normal saline. The brains were carefully extracted from the skulls, and infarct volume was assessed following the method described in our previous work [53]. Briefly, each brain was sectioned into 2-mm thick coronal slices using a brain matrix and then stained with 2% 2,3,5-triphenyltetrazolium chloride (TTC) in normal saline at 37 °C for 30 min. Images of the TTC-stained sections were captured, and the infarcted areas were quantified using ImageJ^®^ software (version 1.53e, National Institutes of Health) based on the specified formula.$$\mathrm{Infarct}\;\mathrm{volume}\;\left(\%\right)=\frac{\left[\mathrm{contralateral}\;\mathrm{hemisphere}\;\mathrm{volume}\right]-\left[\mathrm{non}-\mathrm{infarct}\;\mathrm{ipsilateral}\;\mathrm{hemisphere}\;\mathrm{volume}\right]\;\times \;100}{\left[\mathrm{contralateral}\;\mathrm{hemisphere}\;\mathrm{volume}\right]}$$

### 4.6. Assessment of Neuronal Density

Serial coronal sections (30 µm thick) of the cortex and hippocampus were stained with cresyl violet to assess neuronal density. Specific regions within the cortex and hippocampus (CA1 and CA3) were examined using an Olympus CX23 light microscope (M/S. Magnus Opto Systems India Pvt. Ltd., New Delhi, India). Four coronal sections containing both the cortex and hippocampus were selected based on stereotaxic coordinates, following the method outlined in our previous study [69]. An investigator blinded to the treatment groups examined neuronal density under 40× magnification. Results were expressed as a percentage relative to the control group.

### 4.7. Protein Quantification

Protein levels in the cortex and hippocampus were quantified using the Lowry assay, following established protocols [70]. Calibration curves were generated using bovine serum albumin (MilliporeSigma, Burlington, MA, USA) as the standard. Absorbance was measured at 650 nm using a microplate reader.

### 4.8. Determination of MDA Level

The MDA assay relies on the formation of a colored adduct between malondialdehyde, a marker of lipid peroxidation, and thiobarbituric acid (TBA) when subjected to elevated temperature and acidic conditions. MDA concentrations in brain tissue samples were measured using the thiobarbituric acid (TBA) assay, based on the method described by Ohkawa et al. [71]. Briefly, tissue homogenates were combined with sodium dodecyl sulfate (SDS), acetic acid, and TBA, then incubated at 95 °C for one hour to allow the formation of the MDA-TBA adduct. After cooling, the reaction mixture was extracted with a n-butanol/pyridine solution and centrifuged. The absorbance of the resulting organic phase was read at 532 nm using a spectrophotometer. MDA levels were quantified by comparison with a standard curve generated using 1,1,3,3-tetramethoxypropane (TMP), and results were normalized to total protein content and expressed as nmol/mg protein.

### 4.9. Determination of CAT Activity

The CAT assay measures the enzyme’s capacity to break down hydrogen peroxide (H_2_O_2_) into water and oxygen. The decline in H_2_O_2_ concentration over time serves as an indicator of CAT activity. CAT activity was measured by first homogenizing brain tissue in phosphate buffer while keeping the samples on ice to prevent enzyme degradation. The homogenates were then centrifuged, and the supernatant containing catalase was collected. A reaction mixture was prepared by combining the supernatant with phosphate buffer and H_2_O_2_. The change in absorbance at 240 nanometers was immediately recorded using a spectrophotometer, according to the method described by Goldblith and Proctor [72]. The activity of catalase was expressed as units per milligram of protein (units/mg protein).

### 4.10. Determination of GSH-Px Activity

GSH-Px activity was determined using a commercial assay kit from MilliporeSigma (catalog number: MAK437-1KT). This assay measures the enzyme’s function in catalyzing the reduction of H_2_O_2_ or organic hydroperoxides to water or corresponding alcohols, utilizing reduced glutathione (GSH) as a substrate. During the reaction, GSH is converted to its oxidized form, glutathione disulfide (GSSG), which is then recycled back to GSH by glutathione reductase in the presence of nicotinamide adenine dinucleotide phosphate (NADPH). The consumption of NADPH is tracked by monitoring the decrease in absorbance at 340 nm using a spectrophotometer. Brain tissues from rats were homogenized and centrifuged at 10,000× *g* for 10 min at 4 °C to isolate the enzyme-containing supernatant. This supernatant was combined with phosphate buffer, glutathione reductase, NADPH, and hydrogen peroxide to start the reaction. The rate of NADPH oxidation, reflected by the decline in absorbance, was used to calculate GSH-Px activity, expressed as units/mg protein.

### 4.11. Determination of SOD Activity

The principle of the SOD assay is based on the enzyme’s ability to catalyze the dismutation of superoxide radicals (O_2_^−^) into oxygen and hydrogen peroxide, thereby reducing the concentration of superoxide in the reaction mixture. In this colorimetric assay, SOD activity in the cortex and hippocampus was assessed using the Sigma-Aldrich kit (19160-1K-F) (Sigma-Aldrich, St. Louis, MO, USA), following the manufacturer’s protocol for reagent preparation. Samples were appropriately diluted, and controls were set up as required. In a 96-well plate, specified volumes of the supplied substrate were dispensed, followed by the addition of reaction buffer, then test samples, standards, or controls. The reaction was started by adding the detection reagent, gently mixing, and incubating the plate at 37 °C for 20 min. Absorbance was then recorded at 450 nm using a microplate reader. Each measurement was conducted in duplicate, and the results were reported as units/mg protein.

### 4.12. Western Blot Analysis

At the end of the study, COX-2 and IL-6 protein levels in the cortex and hippocampus were assessed using western blotting, following the method previously described in our earlier work [64]. Briefly, equal amounts of protein were separated on 10% SDS–polyacrylamide gels and transferred onto PVDF membranes (Hybond-P; GE Healthcare Limited, Amersham Place, Little Chalfont, Buckinghamshire, UK). To block nonspecific binding, membranes were incubated with 5% non-fat dry milk in TBS-T (0.1% Tween-20 in Tris-buffered saline, pH 7.4) for 1 h at room temperature. After blocking, membranes were incubated overnight at 4 °C with rabbit monoclonal anti-COX-2 (1:1000), mouse monoclonal anti-IL-6 (1:2000), and rabbit monoclonal anti-β-actin (1:5000) primary antibodies. Following three 5-min washes with TBS-T, membranes were incubated for 1 h at room temperature with either anti-rabbit or anti-mouse IgG secondary antibodies conjugated to peroxidase (1:2000). Protein bands were visualized using a chemiluminescent detection substrate (Supersignal West Pico, Pierce, Rockford, IL, USA). Band intensities for COX-2 and IL-6 were normalized to β-actin, and protein expression levels were quantified using the ChemiDoc™ MP imaging system with Image Lab software (version 6.0.0 build 25, Bio-Rad Laboratories Inc., Hercules, CA, USA).

### 4.13. Statistical Analysis

The results are presented as mean ± standard error of the mean (SEM). Before conducting statistical tests, the normality of the data was evaluated using the Shapiro–Wilk test. Given that the data conformed to a normal distribution, differences among groups were analyzed by one-way analysis of variance (ANOVA), which is suitable for comparing multiple independent groups with normally distributed data. When ANOVA revealed significant differences, Tukey’s post hoc test was applied to perform pairwise comparisons. All statistical analyses were carried out using SPSS^®^ software version 25 (IBM Inc., Chicago, IL, USA), with significance set at *p* < 0.05.

## 5. Conclusions

In conclusion, alpinetin demonstrated significant neuroprotective effects in a rat model of cerebral ischemia by reducing infarct size, preserving neuronal density, and restoring antioxidant enzyme activities. Additionally, alpinetin effectively suppressed the expression of the pro-inflammatory markers COX-2 and IL-6. These findings indicate that alpinetin exerts its protective effects through antioxidant and anti-inflammatory mechanisms and holds promise as a potential therapeutic agent for the treatment of ischemic stroke. Further studies are warranted to elucidate the precise molecular pathways underlying these pharmacological effects.

## Figures and Tables

**Figure 1 ijms-26-05093-f001:**
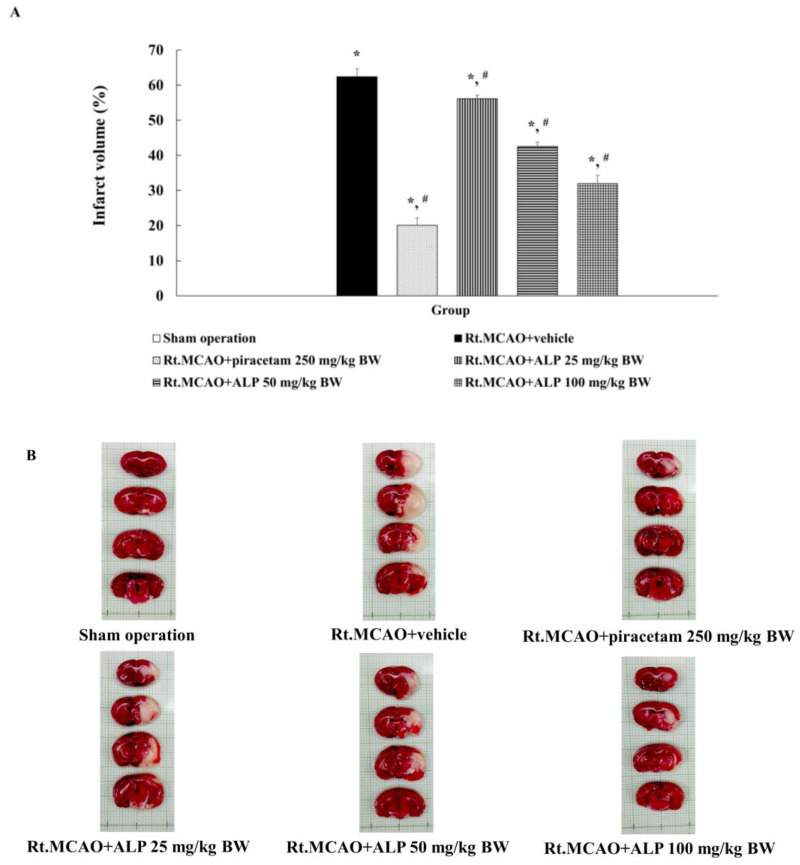
Effect of alpinetin on brain infarct volume. (**A**) Effect of alpinetin on infarct volume in rats with cerebral ischemia. Data are shown as mean ± S.E.M. (n = 5). * *p* < 0.05 compared to the sham operation group; # *p* < 0.05 compared to the Rt.MCAO+vehicle group. (**B**) Representative images of triphenyltetrazolium chloride (TTC)-stained brain sections from each treatment group. Dark staining indicates viable tissue, while the absence of staining denotes infarcted areas. Rt.MCAO, right middle cerebral artery occlusion; BW, body weight; ALP, alpinetin.

**Figure 2 ijms-26-05093-f002:**
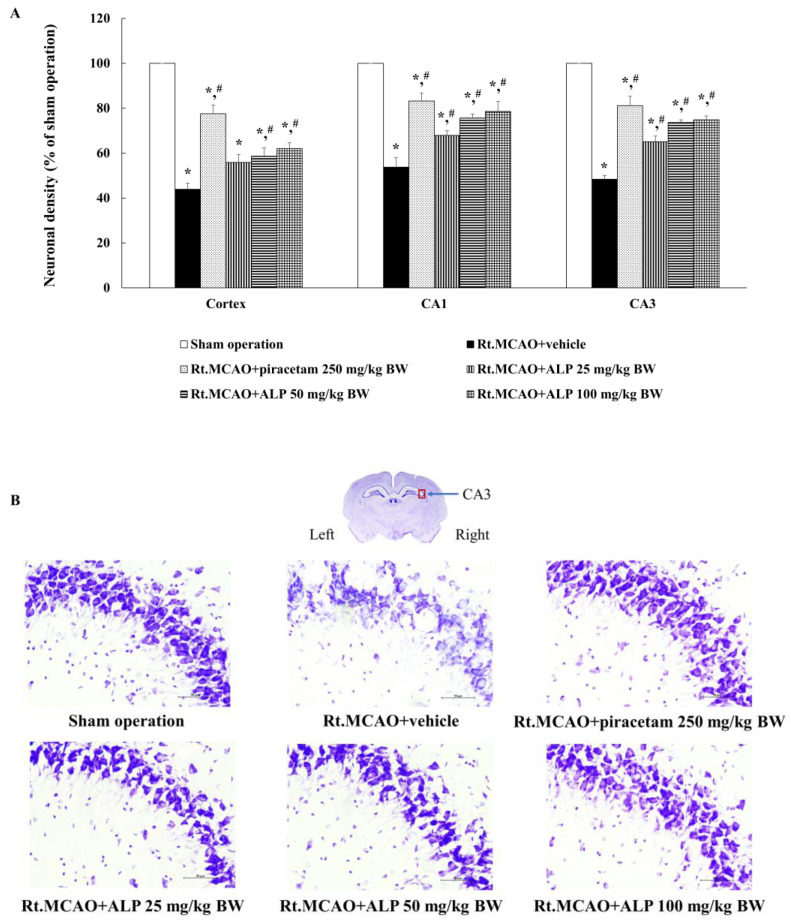
Effect of alpinetin on neuronal density in the cortex and hippocampus. (**A**) A graph depicting neuronal density in the cortex and the CA1 and CA3 regions of the hippocampus. (**B**) Representative images of Nissl-stained coronal sections from the CA3 region of rat brains, taken at 40× magnification. Data are presented as mean ± SEM (n = 5). * *p* < 0.05 compared to the sham operation group; # *p* < 0.05 compared to the Rt.MCAO+vehicle group. Rt.MCAO, right middle cerebral artery occlusion; BW, body weight; ALP, alpinetin.

**Figure 3 ijms-26-05093-f003:**
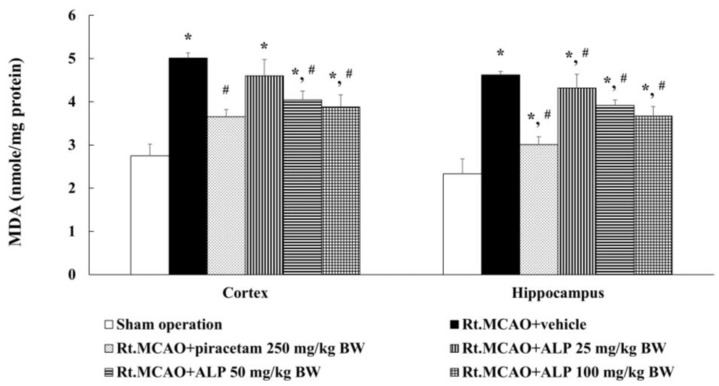
Effect of alpinetin on MDA levels in the cortex and hippocampus. Data are presented as mean ± S.E.M. (n = 5). * *p* < 0.05 compared to the sham-operated group; # *p* < 0.05 compared to the Rt.MCAO+vehicle group. MDA, malondialdehyde; Rt.MCAO, right middle cerebral artery occlusion; BW, body weight; ALP, alpinetin.

**Figure 4 ijms-26-05093-f004:**
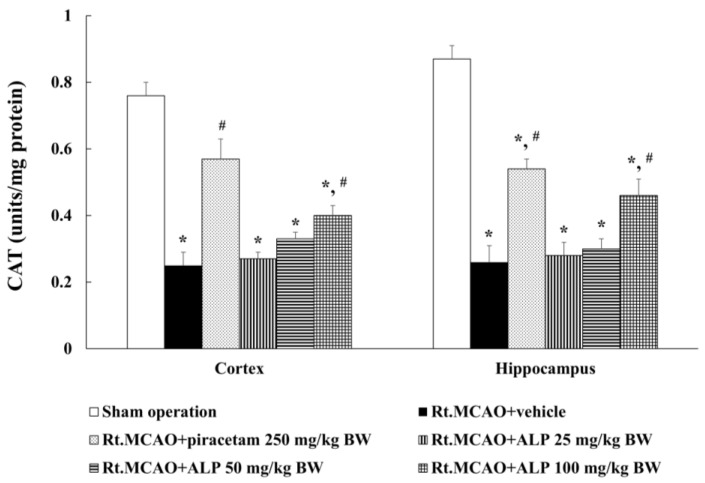
Effect of alpinetin on CAT activity in the cortex and hippocampus. Results are presented as mean ± SEM (n = 5). * *p* < 0.05 compared to the sham operation group; # *p* < 0.05 compared to the Rt.MCAO+vehicle group. CAT, catalase; Rt.MCAO, right middle cerebral artery occlusion; BW, body weight; ALP, alpinetin.

**Figure 5 ijms-26-05093-f005:**
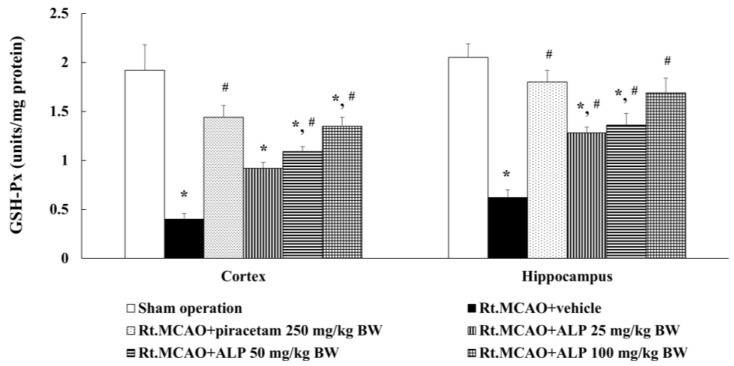
Effects of alpinetin on GSH-Px activity in the cortex and hippocampus. Data are shown as mean ± SEM (n = 5). * *p* < 0.05 versus the sham-operated group; # *p* < 0.05 versus the Rt.MCAO+vehicle group. GSH-Px, glutathione peroxidase; Rt.MCAO, right middle cerebral artery occlusion; BW, body weight; ALP, alpinetin.

**Figure 6 ijms-26-05093-f006:**
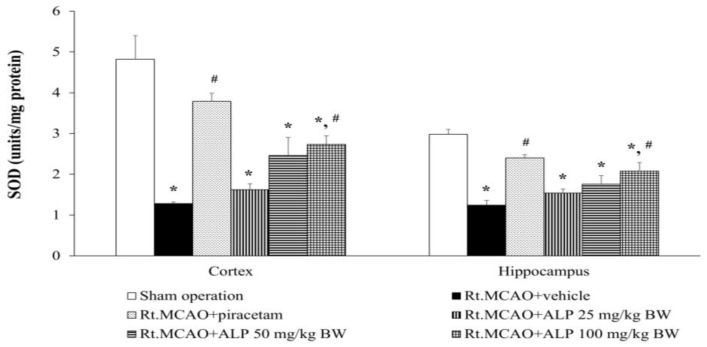
Effects of alpinetin on SOD activity in the cortex and hippocampus. Alpinetin treatment markedly enhanced SOD activity. Data are presented as mean ± SEM (n = 5). * *p* < 0.05 compared to the sham-operated group; # *p* < 0.05 compared to the Rt.MCAO+vehicle group. SOD, superoxide dismutase; Rt.MCAO, right middle cerebral artery occlusion; BW, body weight; ALP, alpinetin.

**Figure 7 ijms-26-05093-f007:**
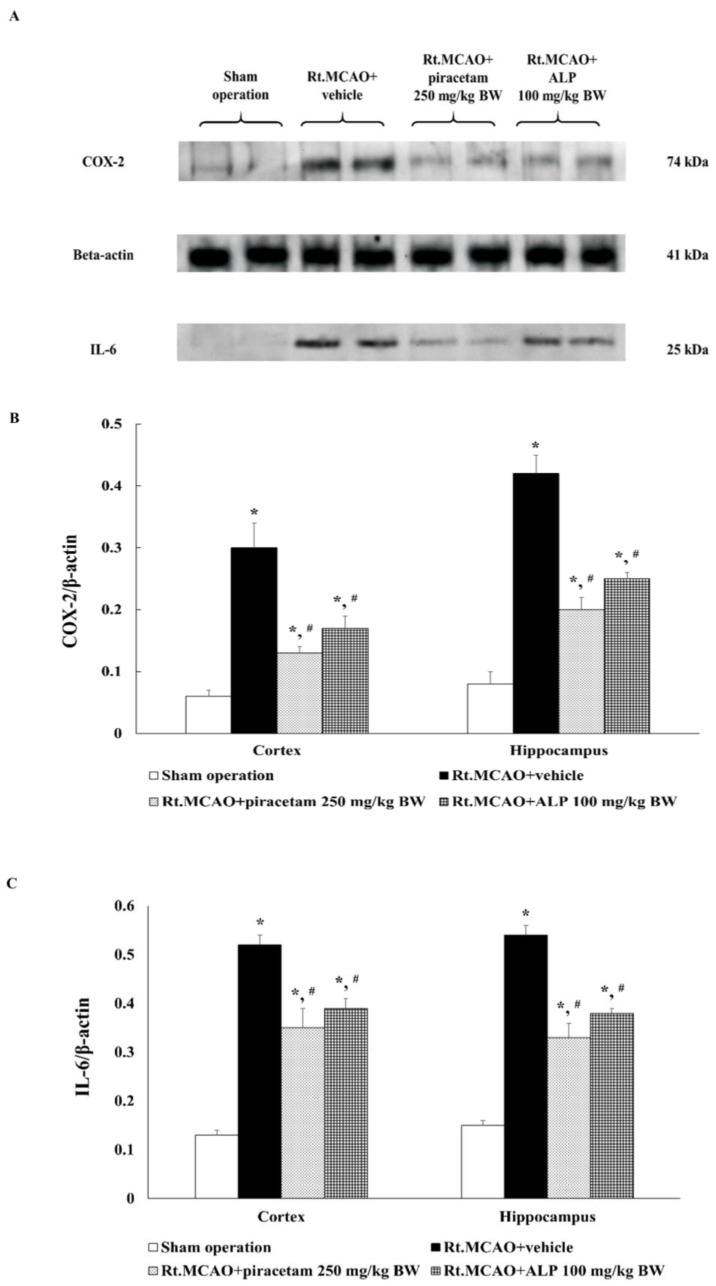
COX-2 and IL-6 expression levels in the cerebral cortex and hippocampus of rats from the sham-operated, Rt.MCAO+vehicle, Rt.MCAO+piracetam (250 mg/kg BW), and Rt.MCAO+ALP (100 mg/kg BW) treatment groups were assessed via Western blot analysis. (**A**) Representative immunoblots showing COX-2 (74 kDa) and IL-6 (25 kDa) expression in the cerebral cortex, with β-actin (41 kDa) serving as the loading control. (**B**) Quantitative analysis of COX-2 expression normalized to β-actin. (**C**) Quantitative analysis of IL-6 expression normalized to β-actin. Data are presented as mean ± S.E.M. (n = 5). * *p* < 0.05 compared to the sham operation group; # *p* < 0.05 compared to the Rt.MCAO+vehicle group. COX-2, cyclooxygenase-2; IL-6, interleukin-6; Rt.MCAO, right middle cerebral artery occlusion; BW, body weight; ALP, alpinetin.

**Figure 8 ijms-26-05093-f008:**
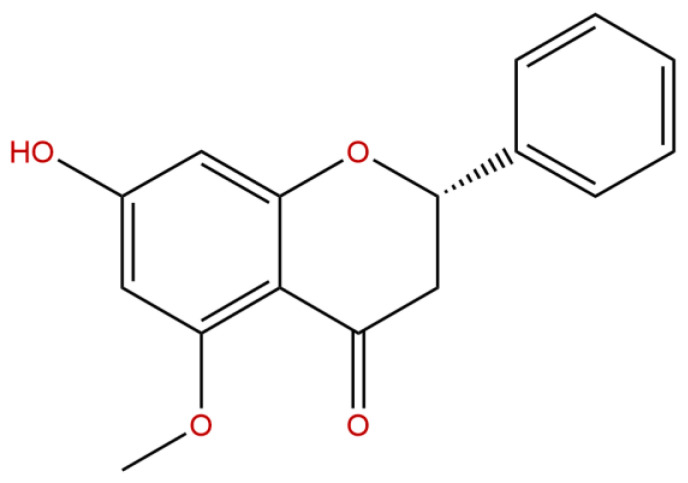
The chemical structure of alpinetin (7-hydroxy-5-methoxyflavanone; C_16_H_14_O_4_).

**Table 1 ijms-26-05093-t001:** Summary of the experimental groups and treatments.

Group Number	Group Name	Treatment	Dose (mg/kg BW)	Route of Administration
1	Sham operation group	-	-	-
2	Rt.MCAO+vehicle-treated group	vehicle	1% DMSO	i.p.
3	Rt.MCAO+piracetam 250 mg/kg BW-treated group	piracetam	250 mg/kg BW	i.p.
4	Rt.MCAO+ALP 25 mg/kg BW-treated group	alpinetin	25 mg/kg BW	i.p.
5	Rt.MCAO+ALP 50 mg/kg BW-treated group	alpinetin	50 mg/kg BW	i.p.
6	Rt.MCAO+ALP 100 mg/kg BW-treated group	alpinetin	100 mg/kg BW	i.p.

## Data Availability

The data presented in this study are available from the corresponding author upon reasonable request.

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
