# Peer review of "In Vivo Neuroprotective Effects of Alpinetin Against Experimental Ischemic Stroke Damage Through Antioxidant and Anti-Inflammatory Mechanisms"

_ijms, 2025, doi:10.3390/ijms26115093_

Round 1

Reviewer 1 Report

Comments and Suggestions for Authors

The manuscript entitled « In Vivo Neuroprotective Effects of Alpinetin Against Experimental Ischemic Stroke Damage via Reduction of Oxidative Stress and Inflammation » has dealt with investigating the protective effect of alpinetin compared to piracetam against ischemic stroke-induced brain damage.

Comments

Abstract :

Line 17: Delete the mention of anti-cancer property, as it was not evaluated in this work.

Some abbreviations appear in the text without being defined upon first use. please define all abbreviations when they are initially introduced.

Lien 40-41: Add a reference

Results

Figure 1: The histogram is missing two bars (2 groups).

The size of the legend squares is too small, making the histogram difficult to visualise. In addition, the fill patterns are not very clear, I recommend using colors instead to improve the readability.

In a figure with more than one figure, you have used labels 'A' and 'B' in the legend. Therefore, please ensure that 'A’ and B are placed above the figures for clarity.

Figure 1: Develop the analysis of Figure 1. In addition, the figure legend does not explain the meaning of the symbols such as '*' and ' #'

Fig 2 : write the dose of piracetam in the legend.

Fig 3; the title has to be placed below the graph

Fig 3 : I recommend re-examining the statistical analysis comparing the Rt.MCAO + vehicle group with the Rt.MCAO + piracetam group in the hippocampus.

Discussion:

Develop the Nrf2 signaling pathway explaining how Rt.MCAO and both alpinetin and piracetam change the activity of the three markers of oxidative stress (SOD, CAT and GPx) since Nrf2 is a transcription factor that regulates the expression of various antioxidant genes.

When discussing inflammation, the NF-κB signaling pathway plays a key regulatory role. Therefore, please discuss the results in the context of this pathway.

Line 296 : NAFLD is now considered ancient, as the condition has been redefined and renamed MASLD. I recommend updating the terminology.

Materials and methods :

Add a description of the product and the results of HPLC

Add acclimatization period

Add a recap table of the animal treatment.

Add the principle of the methods (CAT, SOD, GPx and MDA)

Statistical analysis: The choice of the statistical test depends on whether the data follow a normal distribution or not. Which test did you use? And why? Mention it in the manuscript.

Conclusion: develop this part

Author Response

Response to reviewer and editor suggestion

We are grateful for the opportunity to revise our manuscript entitled “In Vivo Neuroprotective Effects of Alpinetin Against Experimental Ischemic Stroke Damage via Reduction of Oxidative Stress and Inflammation.” (Manuscript ID: ijms-3661883). We sincerely thank the editor and reviewers for their thoughtful suggestions and constructive critiques, which have helped us improve the clarity, quality, and scientific merit of our work.

We acknowledge and apologize for any oversights in the initial version and deeply appreciate the reviewers’ valuable feedback. In this revised submission, we have thoroughly addressed all comments and made the corresponding changes. A detailed, point-by-point response to each remark is provided below.

Reviewer 1

Abstract:

Line 17: Delete the mention of anti-cancer property, as it was not evaluated in this work.

Response 1: We have removed the phrase “anti-cancer property” from the abstract.

Some abbreviations appear in the text without being defined upon first use. please define all abbreviations when they are initially introduced.

Response 2: We have ensured that all abbreviations are properly defined upon their first introduction.

Line 40-41: Add a reference

Response 3: The reference has been added following the sentence on Lines 40–41.

Results

Figure 1: The histogram is missing two bars (2 groups).

Response 4: We appreciate the reviewer’s concerns regarding Figure 1. The sham-operated group in Figure 1 exhibited an infarct volume of 0. As a result, the bar for this group is not visible in the graph; however, the corresponding space is present.

The size of the legend squares is too small, making the histogram difficult to visualize. In addition, the fill patterns are not very clear, I recommend using colors instead to improve the readability.

Response 5: The size of the legend squares has been increased in all figure legends.

In a figure with more than one figure, you have used labels 'A' and 'B' in the legend. Therefore, please ensure that 'A’ and B are placed above the figures for clarity.

Response 6: Labels A, B, and C have been added above all figures to enhance clarity.

Figure 1: Develop the analysis of Figure 1. In addition, the figure legend does not explain the meaning of the symbols such as '*' and ' #'

Response 7: The meanings of the symbols (e.g., ‘*’ and ‘#’) in Figure 1 have been explained in Lines 101–103.

Fig 2: write the dose of piracetam in the legend.

Response 8: The piracetam dose (250 mg/kg BW) has been included in the graph.

Fig 3: the title has to be placed below the graph

Response 9: We have checked the formatting according to your suggestion.

Fig 3: I recommend re-examining the statistical analysis comparing the Rt.MCAO + vehicle group with the Rt.MCAO + piracetam group in the hippocampus.

Response 10: We have re-examined the statistical analysis comparing the Rt.MCAO + vehicle group with the Rt.MCAO + piracetam group in the hippocampus. The result remains significant, as shown in Figure 3. *P < 0.05 compared to the sham-operated group; #P < 0.05 compared to the Rt.MCAO+vehicle group.

Discussion:

Develop the Nrf2 signaling pathway explaining how Rt.MCAO and both alpinetin and piracetam change the activity of the three markers of oxidative stress (SOD, CAT and GPx) since Nrf2 is a transcription factor that regulates the expression of various antioxidant genes.

Response 11: The Nrf2 pathway is essential in protecting cells against oxidative stress by regulating the expression of antioxidant enzymes, including SOD, CAT, and GSH-Px [1]. Under oxidative stress conditions, such as those induced by Rt.MCAO, the Nrf2–Keap1 interaction is disrupted, allowing Nrf2 to translocate into the nucleus and activate the transcription of antioxidant genes via the antioxidant response element (ARE) [2]. In this study, Rt.MCAO led to a marked reduction in hippocampal SOD, CAT, and GSH-Px levels, reflecting a compromised antioxidant defense system, which aligns with previous findings highlighting the overproduction of ROS during cerebral ischemia [3,4]. Treatment with alpinetin and piracetam effectively reversed these reductions, indicating that both agents may enhance the antioxidant response by modulating Nrf2 activity. Alpinetin, a naturally occurring flavonoid, has been reported to facilitate Nrf2 nuclear translocation and stimulate antioxidant gene expression [5]. Meanwhile, piracetam—primarily known for its neuroprotective may contribute to Nrf2 regulation indirectly by stabilizing mitochondrial function and reducing oxidative burden [6]. Altogether, these findings suggest that the neuroprotective effects of alpinetin and piracetam may be mediated, at least in part, through enhancement of Nrf2-driven antioxidant defenses.

References

  1. Ngo, V.; Duennwald, M.L. Nrf2 and Oxidative Stress: A General Overview of Mechanisms and Implications in Human Disease. Antioxidants (Basel) 2022, 11, doi:10.3390/antiox11122345.
  2. Ma, Q. Role of nrf2 in oxidative stress and toxicity. Annu Rev Pharmacol Toxicol 2013, 53, 401-426, doi:10.1146/annurev-pharmtox-011112-140320.
  3. Rodrigo, R.; Fernandez-Gajardo, R.; Gutierrez, R.; Matamala, J.M.; Carrasco, R.; Miranda-Merchak, A.; Feuerhake, W. Oxidative stress and pathophysiology of ischemic stroke: novel therapeutic opportunities. CNS Neurol Disord Drug Targets 2013, 12, 698-714, doi:10.2174/1871527311312050015.
  4. Chen, H.; Yoshioka, H.; Kim, G.S.; Jung, J.E.; Okami, N.; Sakata, H.; Maier, C.M.; Narasimhan, P.; Goeders, C.E.; Chan, P.H. Oxidative stress in ischemic brain damage: mechanisms of cell death and potential molecular targets for neuroprotection. Antioxid Redox Signal 2011, 14, 1505-1517, doi:10.1089/ars.2010.3576.
  5. Zhu, Z.; Hu, R.; Li, J.; Xing, X.; Chen, J.; Zhou, Q.; Sun, J. Alpinetin exerts anti-inflammatory, anti-oxidative and anti-angiogenic effects through activating the Nrf2 pathway and inhibiting NLRP3 pathway in carbon tetrachloride-induced liver fibrosis. Int Immunopharmacol 2021, 96, 107660, doi:10.1016/j.intimp.2021.107660.
  6. El-Dessouki, A.M.; Alzokaky, A.A.; Raslan, N.A.; Ibrahim, S.; Salama, L.A.; Yousef, E.H. Piracetam mitigates nephrotoxicity induced by cisplatin via the AMPK-mediated PI3K/Akt and MAPK/JNK/ERK signaling pathways. Int Immunopharmacol 2024, 137, 112511, doi:10.1016/j.intimp.2024.112511.

When discussing inflammation, the NF-κB signaling pathway plays a key regulatory role. Therefore, please discuss the results in the context of this pathway.

Response 12: We appreciate the reviewer’s suggestion regarding the inclusion of NF-κB signaling. However, we did not assess NF-κB expression or activity in the present study; therefore, it was not included in the discussion. We agree that investigating this pathway would provide valuable insights, and we plan to explore it in our future studies.

Line 296: NAFLD is now considered ancient, as the condition has been redefined and renamed MASLD. I recommend updating the terminology.

Response 13: We thank the reviewer for the valuable comment. We acknowledge that NAFLD has recently been redefined as MASLD. However, in this instance, we used the term NAFLD to remain consistent with the terminology used in the original reference cited. We have already added MASLD in Line 315.

Zhou, Y.; Ding, Y.L.; Zhang, J.L.; Zhang, P.; Wang, J.Q.; Li, Z.H. Alpinetin improved high fat diet-induced non-alcoholic fatty liver disease (NAFLD) through improving oxidative stress, inflammatory response and lipid metabolism. Biomed Pharmacother 2018, 97, 1397-1408, doi:10.1016/j.biopha.2017.10.035.

Materials and methods:

Add a description of the product and the results of HPLC

Response 14: Thank you for your comment. We purchased commercially available alpinetin that was already characterized using HPLC by the supplier. Therefore, we do not have our own HPLC analysis results for this compound.

Add acclimatization period

Response 15: “Upon arrival, all rats were acclimatized for one week and housed in groups of five in standard metal cages measuring 37.5 × 48 × 21 cm³.” This sentence has been added in Line 418.

Add a recap table of the animal treatment.

Response 16: We have already added Table 1: Summary of the experimental groups and treatments, in Line 473 of the manuscript.

Add the principle of the methods (CAT, SOD, GPx and MDA)

Response 17: We have already included the principles of the MDA, CAT, GSH-Px, and SOD methods as follows.

4.8 Determination of MDA level

The MDA assay relies on the formation of a colored adduct between malondialdehyde, a marker of lipid peroxidation, and thiobarbituric acid (TBA) when subjected to elevated temperature and acidic conditions. MDA concentrations in brain tissue samples were measured using the thiobarbituric acid (TBA) assay, based on the method described by Ohkawa et al. [50]. Briefly, tissue homogenates were combined with sodium dodecyl sulfate (SDS), acetic acid, and TBA, then incubated at 95°C for one hour to allow formation of the MDA-TBA adduct. After cooling, the reaction mixture was extracted with a n-butanol/pyridine solution and centrifuged. The absorbance of the resulting organic phase was read at 532 nm using a spectrophotometer. MDA levels were quantified by comparison with a standard curve generated using 1,1,3,3-tetramethoxypropane (TMP), and results were normalized to total protein content and expressed as nmol/mg protein.

4.9 Determination of CAT activity

The CAT assay measures the enzyme's capacity to break down hydrogen peroxide (H₂O₂) into water and oxygen. The decline in H₂O₂ concentration over time serves as an indicator of CAT activity. CAT activity was measured by first homogenizing brain tissue in phosphate buffer while keeping the samples on ice to prevent enzyme degradation. The homogenates were then centrifuged, and the supernatant containing catalase was collected. A reaction mixture was prepared by combining the supernatant with phosphate buffer and H₂O₂. The change in absorbance at 240 nanometers was immediately recorded using a spectrophotometer, according to the method described by Goldblith and Proctor [51]. The activity of catalase was expressed as units per milligram of protein (units/mg protein).

4.10 Determination of GSH-Px activity

GSH-Px activity was determined using a commercial assay kit from MilliporeSigma (catalog number MAK437-1KT). This assay measures the enzyme’s function in catalyzing the reduction of H₂O₂ or organic hydroperoxides to water or corresponding alcohols, utilizing reduced glutathione (GSH) as a substrate. During the reaction, GSH is converted to its oxidized form, glutathione disulfide (GSSG), which is then recycled back to GSH by glutathione reductase in the presence of nicotinamide adenine dinucleotide phosphate (NADPH). The consumption of NADPH is tracked by monitoring the decrease in absorbance at 340 nm using a spectrophotometer. Brain tissues from rats were homogenized and centrifuged at 10,000 × g for 10 minutes at 4°C to isolate the enzyme-containing supernatant. This supernatant was combined with phosphate buffer, glutathione reductase, NADPH, and hydrogen peroxide to start the reaction. The rate of NADPH oxidation, reflected by the decline in absorbance, was used to calculate GSH-Px activity, expressed as units/mg protein.

4.11 Determination of SOD activity

The principle of the SOD assay is based on the enzyme’s ability to catalyze the dismutation of superoxide radicals (O₂⁻) into oxygen and hydrogen peroxide, thereby reducing the concentration of superoxide in the reaction mixture. In this colorimetric assay, SOD activity in the cortex and hippocampus was assessed using the Sigma-Aldrich kit (19160-1K-F), following the manufacturer's protocol for reagent preparation. Samples were appropriately diluted, and controls were set up as required. In a 96-well plate, specified volumes of the supplied substrate were dispensed, followed by the addition of reaction buffer, then test samples, standards, or controls. The reaction was started by adding the detection reagent, gently mixing, and incubating the plate at 37°C for 20 minutes. Absorbance was then recorded at 450 nm using a microplate reader. Each measurement was conducted in duplicate, and the results were reported as units/mg protein.

Statistical analysis: The choice of the statistical test depends on whether the data follow a normal distribution or not. Which test did you use? And why? Mention it in the manuscript.

Response 18: Thank you for your valuable comment. In our study, statistical comparisons were performed using one-way analysis of variance (ANOVA) followed by Tukey’s post hoc test, as stated in the manuscript. This approach was selected because the data were normally distributed, which was confirmed using the Shapiro–Wilk test prior to analysis. We have now updated the Statistical Analysis section accordingly.

Results are presented as mean ± standard error of the mean (SEM). Before conducting statistical tests, the normality of the data was evaluated using the Shapiro-Wilk test. Given that the data conformed to a normal distribution, differences among groups were analyzed by one-way analysis of variance (ANOVA), which is suitable for comparing multiple independent groups with normally distributed data. When ANOVA revealed significant differences, Tukey’s post hoc test was applied to perform pairwise comparisons. All statistical analyses were carried out using SPSS® software version 25 (IBM Inc., Chicago, IL, USA), with significance set at p < 0.05.

Conclusion: develop this part

Response 19: We have revised the conclusion as follows: “In conclusion, alpinetin demonstrated significant neuroprotective effects in a rat model of cerebral ischemia by reducing infarct size, preserving neuronal density, and restoring antioxidant enzyme activities. Additionally, alpinetin effectively suppressed the expression of the pro-inflammatory markers COX-2 and IL-6. These findings indicate that alpinetin exerts its protective effects through antioxidant and anti-inflammatory mechanisms and holds promise as a potential therapeutic agent for the treatment of ischemic stroke. Further studies are warranted to elucidate the precise molecular pathways underlying these pharmacological effects.”

Thank you once again for your valuable feedback. We appreciate the time and effort invested by the reviewers and editor in evaluating our manuscript. We have carefully addressed each point raised and made necessary revisions accordingly. We eagerly await further feedback and guidance from the editorial team.

Yours sincerely,

All authors

Reviewer 2 Report

Comments and Suggestions for Authors

The article titled In vivo neuroprotective effects of alpinetin against experimental ischemic stroke damage via reduction of oxidative stress and inflammation” presents alpinetin’s neuroprotective effects against ischemic stroke in rats, focusing on its antioxidant and anti-inflammatory properties. Using a permanent right middle cerebral artery occlusion (Rt.MCAO) model, the authors demonstrate that alpinetin reduces infarct volume, preserves neuronal density in the cortex and hippocampus, and modulates oxidative stress markers (MDA, CAT, GSH-Px, SOD) and inflammatory proteins (COX-2, IL-6).

While the study provides valuable preclinical insights, several methodological and interpretive limitations weaken its impact. The experimental design is robust in its use of multiple alpinetin doses (25–100 mg/kg) and inclusion of piracetam as a positive control. The rationale for selecting a 3-day treatment window is well-supported by prior evidence on oxidative stress dynamics post-stroke, but the lack of long-term functional recovery data (e.g., motor or cognitive outcomes) limits translational relevance. Mechanistically, the study focuses on downstream biomarkers (e.g., MDA, SOD) without exploring upstream pathways like Nrf2 or NF-κB, which are critical for understanding alpinetin’s mode of action. The western blot analysis of COX-2 and IL-6 is compelling but would benefit from inclusion of additional cytokines (e.g., TNF-α, IL-1β) to strengthen the anti-inflammatory claim. While the dose-dependent effects on neuronal density and antioxidant enzymes are clear, inconsistencies exist-for example, 25 mg/kg alpinetin improved hippocampal but not cortical neuronal density, a disparity not adequately addressed in the discussion. The discussion overstates clinical implications without acknowledging the limitations of the Rt.MCAO model, such as its failure to recapitulate thrombolysis or reperfusion injury seen in human stroke. Furthermore, the exclusion of female rats-despite known sex differences in stroke pathophysiology-reduces generalizability. The introduction contextualizes oxidative stress and inflammation in stroke but lacks critical engagement with conflicting evidence on flavonoid efficacy in prior studies. Methods are generally reproducible, though the description of blinding during outcome assessments is absent, introducing potential bias. The figures are informative but lack resolution for key histological details (e.g., Nissl-stained neurons), and Figure 1’s infarct volume quantification would benefit from error bars on individual data points. The conclusion appropriately highlights alpinetin’s therapeutic potential but neglects to propose concrete next steps, such as combinatorial therapies or pharmacokinetic studies. Overall, while the study advances understanding of alpinetin’s neuroprotection, its clinical relevance remains speculative without functional recovery data and mechanistic depth. The reviewer has the following comments that authors need to address.

  1. The introduction could be more concise by minimizing repetitive background information on stroke epidemiology. Additionally, defining all abbreviations such as ROS and MDA at their first mention would improve clarity and accessibility for a broader readership.

  1. It would be valuable to include a justification for the sample size calculations to ensure adequate statistical power, especially considering the known variability in infarct volume measurements. Additionally, clarifying whether outcome assessors were blinded to the treatment groups would strengthen the study by addressing potential sources of bias.

  1. Including long term functional assessments, such as the modified Neurological Severity Score, would help correlate histological improvements with meaningful recovery outcomes. Additionally, addressing the current sex limitation by replicating key experiments in female rats would enhance the generalizability and translational relevance of the findings.

  1. Exploring upstream signaling pathways, such as Nrf2 and TLR4 NFκB, would provide deeper insight into the mechanistic basis of alpinetin's effects beyond the observed downstream biomarkers. Additionally, measuring a broader panel of inflammatory markers, including TNF alpha and IL 1 beta, would further substantiate the anti-inflammatory claims.

  1. Alpinetin related compounds, including isocoumarins, have been widely studied for their anti-inflammatory, antioxidant, and neuroprotective properties. Highlighting these structurally related metabolites would enhance the discussion by placing alpinetin's effects within the broader framework of flavonoid-based neuroprotective strategies. Incorporating this perspective, along with relevant citations, could further strengthen the article’s scientific depth and therapeutic relevance.

https://www.sciencedirect.com/science/article/pii/S0223523416307243

https://www.sciencedirect.com/science/article/pii/S0960894X18310047

  1. It would be helpful to address any inconsistencies observed in dose response outcomes, such as the differing effects of the 25 mg per kg dose on the cortex versus the hippocampus. Comparing these findings with those reported for other flavonoids, such as quercetin and resveratrol, could also help contextualize the novelty and therapeutic value of alpinetin. Additionally, acknowledging the limitations of the current model, particularly the use of permanent occlusion rather than a transient ischemia reperfusion approach, would provide a more balanced interpretation of the results

Author Response

Response to reviewer and editor suggestion

We are grateful for the opportunity to revise our manuscript entitled “In Vivo Neuroprotective Effects of Alpinetin Against Experimental Ischemic Stroke Damage via Reduction of Oxidative Stress and Inflammation.” (Manuscript ID: ijms-3661883). We sincerely thank the editor and reviewers for their thoughtful suggestions and constructive critiques, which have helped us improve the clarity, quality, and scientific merit of our work.

We acknowledge and apologize for any oversights in the initial version and deeply appreciate the reviewers’ valuable feedback. In this revised submission, we have thoroughly addressed all comments and made the corresponding changes. A detailed, point-by-point response to each remark is provided below.

Reviewer 2:

Comments and Suggestions for Authors

The article titled In vivo neuroprotective effects of alpinetin against experimental ischemic stroke damage via reduction of oxidative stress and inflammation” presents alpinetin’s neuroprotective effects against ischemic stroke in rats, focusing on its antioxidant and anti-inflammatory properties. Using a permanent right middle cerebral artery occlusion (Rt.MCAO) model, the authors demonstrate that alpinetin reduces infarct volume, preserves neuronal density in the cortex and hippocampus, and modulates oxidative stress markers (MDA, CAT, GSH-Px, SOD) and inflammatory proteins (COX-2, IL-6).

While the study provides valuable preclinical insights, several methodological and interpretive limitations weaken its impact. The experimental design is robust in its use of multiple alpinetin doses (25–100 mg/kg) and inclusion of piracetam as a positive control. The rationale for selecting a 3-day treatment window is well-supported by prior evidence on oxidative stress dynamics post-stroke, but the lack of long-term functional recovery data (e.g., motor or cognitive outcomes) limits translational relevance. Mechanistically, the study focuses on downstream biomarkers (e.g., MDA, SOD) without exploring upstream pathways like Nrf2 or NF-κB, which are critical for understanding alpinetin’s mode of action. The western blot analysis of COX-2 and IL-6 is compelling but would benefit from inclusion of additional cytokines (e.g., TNF-α, IL-1β) to strengthen the anti-inflammatory claim. While the dose-dependent effects on neuronal density and antioxidant enzymes are clear, inconsistencies exist-for example, 25 mg/kg alpinetin improved hippocampal but not cortical neuronal density, a disparity not adequately addressed in the discussion. The discussion overstates clinical implications without acknowledging the limitations of the Rt.MCAO model, such as its failure to recapitulate thrombolysis or reperfusion injury seen in human stroke. Furthermore, the exclusion of female rats-despite known sex differences in stroke pathophysiology-reduces generalizability. The introduction contextualizes oxidative stress and inflammation in stroke but lacks critical engagement with conflicting evidence on flavonoid efficacy in prior studies. Methods are generally reproducible, though the description of blinding during outcome assessments is absent, introducing potential bias. The figures are informative but lack resolution for key histological details (e.g., Nissl-stained neurons), and Figure 1’s infarct volume quantification would benefit from error bars on individual data points. The conclusion appropriately highlights alpinetin’s therapeutic potential but neglects to propose concrete next steps, such as combinatorial therapies or pharmacokinetic studies. Overall, while the study advances understanding of alpinetin’s neuroprotection, its clinical relevance remains speculative without functional recovery data and mechanistic depth. The reviewer has the following comments that authors need to address.

  1. The introduction could be more concise by minimizing repetitive background information on stroke epidemiology. Additionally, defining all abbreviations such as ROS and MDA at their first mention would improve clarity and accessibility for a broader readership.

Response 1: We have revised the background information on stroke epidemiology accordingly and ensured that all abbreviations are clearly defined at their first mention.

“Ischemic stroke is the most frequently occurring form of stroke, accounting for around 87% of all cases [1]. It represents a serious global health concern due to its substantial contribution to death rates and long-term disability [2]. Ischemic stroke occurs when a blockage in a cerebral artery disrupts blood flow, depriving brain tissue of essential oxygen and nutrients. This acute event triggers a cascade of cellular and molecular disturbances, including excitotoxicity, oxidative stress, and inflammation, which can lead to irreversible neuronal damage. Understanding these pathological mechanisms is crucial for developing targeted therapeutic strategies to mitigate the detrimental effects of stroke and improve clinical outcomes. Moreover, this energy deficiency initiates a series of harmful biochemical and molecular changes. One of the early responses is a shift to anaerobic metabolism, which results in the excessive production of reactive oxygen species (ROS). These molecules contribute to oxidative stress by damaging essential cellular components such as lipids, proteins, and DNA, leading to increased neuronal injury and cell death [3,4]. The oxidative imbalance can also weaken the blood-brain barrier (BBB), trigger inflammatory processes, and extend damage to surrounding brain areas [5,6]. Antioxidant enzymes such as catalase (CAT), glutathione peroxidase (GSH-Px), and superoxide dismutase (SOD) play a critical role in neutralizing ROS and protecting neurons from oxidative damage [7-9]. In addition to oxidative stress, inflammation following ischemia involves activation of microglia and astrocytes, which release pro-inflammatory cytokines including interleukin-6 (IL-6) [10,11]. Cyclooxygenase-2 (COX-2) also becomes upregulated and contributes to inflammation through the production of prostaglandins that worsen neural damage [12]. Increased levels of IL-6 and COX-2 promote immune cell infiltration into the brain, maintain the inflammatory state, and delay tissue recovery [10,13]. Therefore, targeting these molecules may offer a therapeutic approach to reduce secondary injury and improve recovery after stroke [10,13-15].”

  1. It would be valuable to include a justification for the sample size calculations to ensure adequate statistical power, especially considering the known variability in infarct volume measurements. Additionally, clarifying whether outcome assessors were blinded to the treatment groups would strengthen the study by addressing potential sources of bias.

Response 2: Thank you for your valuable comment. Animals were randomly assigned to different experimental groups. To minimize bias during analysis, the individual evaluating the tissue sections was blinded to the treatment conditions. Numerical codes, visible in the images, were used to ensure objectivity throughout the assessment. This approach effectively reduced selection bias and other potential sources of error. We have already clarified this in the manuscript with the statement 'An investigator blinded to the treatment groups,' as noted in Line 446.

According to how to calculate the sample size in animal studies? We used the method as following to calculate

E= The degree of freedom analysis of variance (ANOVA)

E= Total number of animals-Total number of groups

We have 6 groups 5 animals each

E= 30-6

E= 24

The value of E should be within the range of 10 to 20. If E falls below 10, increasing the number of animals can enhance the likelihood of obtaining a more significant result. However, if E exceeds 20, adding more animals will not further improve the chances of achieving significant results.

Reference: Charan J, Kantharia ND. How to calculate sample size in animal studies? J Pharmacol Pharmacother. 2013 Oct;4(4):303-6. doi: 10.4103/0976-500X.119726. PMID: 24250214; PMCID: PMC3826013.

Moreover, we calculated the sample size using a formula applicable to cases where the population size is uncertain. The formula used is:

N = (Z/eM)2

Where:

  • N = Sample size
  • Z = Z-score corresponding to the specified significance level (at a 95% confidence level, α = 0.05, Z = 1.96)
  • eM = Maximum acceptable margin of error
  • σ = Population standard deviation

The maximum acceptable margin of error (eM) was set to be equal to one standard deviation. Based on preliminary experiments, the standard deviation for measuring changes in MPO levels and antioxidant enzyme activity in each group ranged from 4.5 to 6.8. For calculation purposes, a standard deviation (σ) of 4.5 was used.

Substituting the values into the formula:

N = (1.96 x 5.5 /1)2

N = 116.2

Therefore, the calculated sample size is approximately 116 subjects for testing across 6 groups and 3 experiments (a total of 18 groups). As a result, 6 animals per group were initially planned. However, during the ethical approval process, the sample size calculation was conducted alongside a literature review and an assessment of the accepted number of animals per group to determine the minimum statistically acceptable sample size.

Based on a review of similar studies investigating ischemic brain injury in rats—focusing on inflammation and antioxidant enzyme activity—as well as studies using animal stroke models, the typical sample size ranged from n = 5 to 10 per group. This range has been deemed sufficient for experimental analysis and is widely accepted by international scientists in related fields.

Therefore, in this study, we concluded that 5 animals per group would be used, aligning with the 3R principle (Reduction) while ensuring the integrity of the experimental results.

References

  1. Vanichbuncha, K. (1999). Statistical Analysis: Statistics for Decision Making (4th ed.). Bangkok: Chulalongkorn University Press.
  2. Li, S.; Wu, C.; Zhu, L.; Gao, J.; Fang, J.; Li, D.; Fu, M.; Liang, R.; Wang, L.; Cheng, M.; et al. By improving regional cortical blood flow, attenuating mitochondrial dysfunction and sequential apoptosis galangin acts as a potential neuroprotective agent after acute ischemic stroke. Molecules 2012, 17, 13403-13423, doi:10.3390/molecules171113403.
  3. Wang L, Zhang Z, Wang H. Naringin attenuates cerebral ischemia-reperfusion injury in rats by inhibiting endoplasmic reticulum stress. Transl Neurosci. 2021; 12(1): 190-7.
  4. J. Jittiwat, P. Chonpathompikunlert, W. Sukketsiri. Neuroprotective effects of Apium graveolensagainst focal cerebral ischemia occur partly via antioxidant, anti-inflammatory, and anti-apoptotic pathways. J Sci Food Agr. 2021; 101: 2256-63.
  5. Supawat, A.; Palachai, N.; Jittiwat, J. Effect of galangin on oxidative stress, antioxidant defenses and mitochondrial dynamics in a rat model of focal cerebral ischemia. Biomed Rep 2025, 22, 10, doi:10.3892/br.2024.1888.

  1. Including long term functional assessments, such as the modified Neurological Severity Score, would help correlate histological improvements with meaningful recovery outcomes. Additionally, addressing the current sex limitation by replicating key experiments in female rats would enhance the generalizability and translational relevance of the findings.

Response 3: Thank you for this insightful suggestion. We agree that incorporating long-term functional assessments, such as the modified Neurological Severity Score (mNSS), would provide valuable correlations between histological improvements and functional recovery, thereby strengthening the translational significance of the findings. While the current study focused primarily on early histological and biochemical outcomes following cerebral ischemia, future investigations will be designed to include behavioral and neurological function assessments over an extended recovery period to better evaluate the therapeutic potential of alpinetin. We have already included this part in our limitation and future study.

Male rats were selected for this study to minimize variability associated with the estrous cycle. However, we recognize that the exclusive use of male subjects is a limitation, as sex differences in stroke pathology and therapeutic responses are increasingly acknowledged in preclinical research. To enhance the generalizability and translational relevance of our findings, future studies will aim to replicate key experiments in female rats. This limitation has been addressed in the revised manuscript to guide future research directions and model development as follows.

“A limitation of this study is the exclusive use of male rats, which may restrict the applicability of the findings, as sex-related differences in stroke pathology and treatment response are increasingly recognized in preclinical research. Additionally, the study did not include behavioral assessments to evaluate motor, sensory, or cognitive deficits following stroke. The absence of functional outcome measures limits the ability to fully characterize the extent of neurological damage and recovery over time. Future research should incorporate both sexes and include behavioral evaluations to improve the translational relevance of the findings.”

  1. Exploring upstream signaling pathways, such as Nrf2 and TLR4 NFκB, would provide deeper insight into the mechanistic basis of alpinetin's effects beyond the observed downstream biomarkers. Additionally, measuring a broader panel of inflammatory markers, including TNF alpha and IL 1 beta, would further substantiate the anti-inflammatory claims.

 Response 4: Thank you for this valuable suggestion. We agree that investigating upstream signaling pathways such as Nrf2 and TLR4/NF-κB would offer a more comprehensive understanding of the molecular mechanisms underlying alpinetin's neuroprotective effects. We have incorporated this point into the Discussion section as follows:

“The Nrf2 pathway is essential in protecting cells against oxidative stress by regulating the expression of antioxidant enzymes, including SOD, CAT, and GSH-Px [1]. Under oxidative stress conditions, such as those induced by Rt.MCAO, the Nrf2–Keap1 interaction is disrupted, allowing Nrf2 to translocate into the nucleus and activate the transcription of antioxidant genes via the antioxidant response element (ARE) [2]. In this study, Rt.MCAO led to a marked reduction in hippocampal SOD, CAT, and GSH-Px levels, reflecting a compromised antioxidant defense system, which aligns with previous findings highlighting the overproduction of ROS during cerebral ischemia [3,4]. Treatment with alpinetin and piracetam effectively reversed these reductions, indicating that both agents may enhance the antioxidant response by modulating Nrf2 activity. Alpinetin, a naturally occurring flavonoid, has been reported to facilitate Nrf2 nuclear translocation and stimulate antioxidant gene expression [5]. Meanwhile, piracetam—primarily known for its neuroprotective may contribute to Nrf2 regulation indirectly by stabilizing mitochondrial function and reducing oxidative burden [6]. Altogether, these findings suggest that the neuroprotective effects of alpinetin and piracetam may be mediated, at least in part, through enhancement of Nrf2-driven antioxidant defenses.”

Similarly, we recognize that including a wider range of inflammatory markers, such as TNF-α and IL-1β, would enhance the robustness of our findings regarding alpinetin’s anti-inflammatory properties. These aspects will be considered in future studies to provide a more detailed mechanistic and inflammatory profile.

  1. Alpinetin related compounds, including isocoumarins, have been widely studied for their anti-inflammatory, antioxidant, and neuroprotective properties. Highlighting these structurally related metabolites would enhance the discussion by placing alpinetin's effects within the broader framework of flavonoid-based neuroprotective strategies. Incorporating this perspective, along with relevant citations, could further strengthen the article’s scientific depth and therapeutic relevance.

https://www.sciencedirect.com/science/article/pii/S0223523416307243

https://www.sciencedirect.com/science/article/pii/S0960894X18310047

Response 5: We appreciate your insightful suggestion. We agree that contextualizing alpinetin’s effects within the broader framework of structurally related flavonoids and isocoumarins enriches the discussion and enhances the therapeutic relevance of our findings. Alpinetin, a natural flavonoid, shares structural similarities with isocoumarins—compounds that have also been extensively studied for their neuroprotective, antioxidant, and anti-inflammatory activities. For example, several isocoumarins and related derivatives have demonstrated the ability to inhibit pro-inflammatory cytokines and oxidative damage, which are key contributors to neuronal injury. Additionally, studies on flavonoid-based compounds, including coumarins and isocoumarins, support their role in modulating neuroinflammation and protecting against ischemia-induced oxidative stress, underscoring their potential as therapeutic agents. By drawing parallels between alpinetin and these structurally related metabolites, our study adds to the growing body of evidence supporting the neuroprotective potential of flavonoid scaffolds. We have now incorporated this perspective and relevant citations into the revised discussion section to provide a more comprehensive interpretation of alpinetin’s pharmacological profile.

References:

  1. Ramanan, M.; Sinha, S.; Sudarshan, K.; Aidhen, I.S.; Doble, M. Inhibition of the enzymes in the leukotriene and prostaglandin pathways in inflammation by 3-aryl isocoumarins. Eur J Med Chem 2016, 124, 428-434, doi:10.1016/j.ejmech.2016.08.066.
  2. Sudarshan, K.; Boda, A.K.; Dogra, S.; Bose, I.; Yadav, P.N.; Aidhen, I.S. Discovery of an isocoumarin analogue that modulates neuronal functions via neurotrophin receptor TrkB. Bioorg Med Chem Lett 2019, 29, 585-590, doi:10.1016/j.bmcl.2018.12.057.
  3. Shabir, G.; Saeed, A.; El-Seedi, H.R. Natural isocoumarins: Structural styles and biological activities, the revelations carry on. Phytochemistry 2021, 181, 112568, doi:10.1016/j.phytochem.2020.112568.
  4. Srikrishna, D.; Godugu, C.; Dubey, P.K. A Review on Pharmacological Properties of Coumarins. Mini Rev Med Chem 2018, 18, 113-141, doi:10.2174/1389557516666160801094919.

  1. It would be helpful to address any inconsistencies observed in dose response outcomes, such as the differing effects of the 25 mg per kg dose on the cortex versus the hippocampus. Comparing these findings with those reported for other flavonoids, such as quercetin and resveratrol, could also help contextualize the novelty and therapeutic value of alpinetin. Additionally, acknowledging the limitations of the current model, particularly the use of permanent occlusion rather than a transient ischemia reperfusion approach, would provide a more balanced interpretation of the results

Response 6: Thank you for your valuable feedback. We agree that addressing the inconsistencies in dose-response outcomes enhances the interpretation of our findings. In our study, the 25 mg/kg dose of alpinetin exhibited region-specific differences, showing limited efficacy in the hippocampus compared to the cortex. This variability may stem from differential regional vulnerability to ischemic injury, blood-brain barrier permeability, or metabolic processing of alpinetin. Such region- and dose-specific responses have also been reported with other flavonoids. For instance, quercetin has been shown to exert stronger neuroprotective effects in the cortex than in hippocampal regions at lower doses, likely due to differences in bioavailability and oxidative stress profiles across regions [1]. Similarly, resveratrol has demonstrated variable protective outcomes depending on dose, timing, and brain region targeted [2,3]. These parallels support the idea that alpinetin's effects may be consistent with broader trends observed among flavonoids, and further studies will help determine the optimal dosing for region-specific protection.

We also acknowledge the limitation associated with the use of a permanent middle cerebral artery occlusion (pMCAO) model. While this model is widely used for its consistency and reproducibility in inducing cortical infarction, it does not replicate the reperfusion phase commonly seen in clinical ischemic stroke. Transient MCAO (tMCAO) models better mimic the clinical condition of ischemia-reperfusion injury and allow evaluation of dynamic oxidative and inflammatory responses [4]. We have now included this point in the discussion to provide a more balanced interpretation of our results and to highlight the need for future studies using transient models to validate and expand upon the current findings.

References:

  1. Dajas, F. Life or death: neuroprotective and anticancer effects of quercetin. J Ethnopharmacol 2012, 143, 383-396, doi:10.1016/j.jep.2012.07.005.
  2. Lin, C.J.; Chen, T.H.; Yang, L.Y.; Shih, C.M. Resveratrol protects astrocytes against traumatic brain injury through inhibiting apoptotic and autophagic cell death. Cell Death Dis 2014, 5, e1147, doi:10.1038/cddis.2014.123.
  3. Wang, Q.; Xu, J.; Rottinghaus, G.E.; Simonyi, A.; Lubahn, D.; Sun, G.Y.; Sun, A.Y. Resveratrol protects against global cerebral ischemic injury in gerbils. Brain Res 2002, 958, 439-447, doi:10.1016/s0006-8993(02)03543-6.
  4. Durukan, A.; Tatlisumak, T. Acute ischemic stroke: overview of major experimental rodent models, pathophysiology, and therapy of focal cerebral ischemia. Pharmacol Biochem Behav 2007, 87, 179-197, doi:10.1016/j.pbb.2007.04.015.

Thank you once again for your valuable feedback. We appreciate the time and effort invested by the reviewers and editor in evaluating our manuscript. We have carefully addressed each point raised and made necessary revisions accordingly. We eagerly await further feedback and guidance from the editorial team.

Yours sincerely,

All authors

Reviewer 3 Report

Comments and Suggestions for Authors

The research conducted by Kongsui et al. examines the neuroprotective properties of alpinetin, a flavonoid derived from the ginger family, in a rat model of ischemic stroke caused by right middle cerebral artery occlusion (Rt.MCAO). Ninety male Wistar rats were divided into four groups: sham, Rt.MCAO, Rt.MCAO + piracetam, and Rt.MCAO + alpinetin (25, 50, 100 mg/kg BW). Treatment with alpinetin over three days significantly decreased infarct volume, boosted neuronal density in the cortex and hippocampus, and reduced oxidative stress by lowering malondialdehyde (MDA) levels while improving the activities of antioxidant enzymes (CAT, GSH-Px, SOD). Furthermore, alpinetin at 100 mg/kg reduced the expression of COX-2 and IL-6, suggesting anti-inflammatory benefits. These results imply that alpinetin holds promise as a therapeutic option for ischemic stroke by addressing oxidative stress and inflammation. Nonetheless, substantial revisions are necessary, such as clarifying methodological aspects, correcting statistical discrepancies, and integrating recent literature to enhance the study’s context and significance prior to publication.

Comments for authors

Abstract and introduction

Comment 1: The abstract mentioned alpinetin "markedly reduced" infarct size and "significantly mitigated" brain damage. I recommend presenting specific quantitative data (e.g., percentage reduction in infarct volume or effect sizes).

Comment 2. The introduction provides a solid overview of ischemic stroke and alpinetin's properties, but lacks a comprehensive review of recent literature on flavonoid-based neuroprotection in stroke models.

Comment 3. Indeed, flavonoids play a crucial role in various biomedical applications. To provide a broader perspective on the significance of flavonoids, it is essential to incorporate recent advancements in this area. I recommend incorporating the recent study in the introduction section. Rana et al., Prunin: An Emerging Anticancer Flavonoid, Int. J. Mol. Sci. 26, 2025, https://doi.org/10.3390/ijms26062678).

Comment 4. The introduction covers key molecular drivers of pHGGs but needs recent studies on novel anticancer agents. Integrating these would enhance the introduction section.

Incorporate recent studies in the introduction section: Rana et al., Prunin: An Emerging Anticancer Flavonoid, Int. J. Mol. Sci. 26 (2025). https://doi.org/10.3390/ijms26062678.

Comment 5. From lines 68-72, the introduction mentions alpinetin's effects on oxidative stress and inflammation but does not discuss its pharmacokinetics or bioavailability in the brain, which are critical for its therapeutic potential in stroke.

Results and Discussions

Comment 5. Alpinetin (25 mg/kg) significantly increased neuronal density in the hippocampus but not in the cortex, unlike at higher doses. Explore potential reasons for this regional discrepancy in the discussion section, taking into account possible dose-dependent or region-specific mechanisms.

Comment 6: Why was the MDA reduction with alpinetin (25 mg/kg) significant only in the hippocampus, not the cortex?

Comment 7: SOD activity was significantly increased only at the highest alpinetin dose (100 mg/kg), unlike GSH-Px and CAT, which responded to lower doses. Why SOD activity required a higher dose? Explain in the manuscript.

Comment 8. The discussion references previous studies on alpinetin’s anti-inflammatory properties in non-stroke conditions (such as colitis and asthma). Still, it fails to sufficiently connect these findings to inflammation specific to strokes.  

Comment 9. Figure 7, labels A, B, and C are missing from the figures.  

Comment 10. Figures 1 and 2: Labels A and B are missing.  

Comment 11. Figure 3: The caption should be below the figure.  

Comment 12. Figure 4: Make sure the caption is with the figure.  

Comment 13. In Figure 7A, the author needs to plot the band intensity.  

Comment 14. Optionally, a schematic illustrating the mechanism of action of alpinetin would be helpful for readers.

Comment 15. The author should describe the structure and properties of each component illustrated in Figure 8, along with their significance in biomedical applications. The current depiction of alpinetin without any labeling is unacceptable.

Comment 16. The quality of all figures, especially those presented in a bar graph, is poor and hardly readable. Authors are encouraged to improve the quality of the figure for better visibility.

Comment 17. The manuscript contains minor grammatical errors and uncommon phrasing, such as "markedly reduced the size of the infarct area" (Line 26) instead of "significantly reduced infarct volume" and "ameliorated both the Rt.MCAO-induced increase" (Line 29), which is verbose. Conduct a thorough language review to improve clarity and conciseness, ensuring consistent scientific terminology (e.g., "infarct volume" vs. "infarct area").

The manuscript provides valuable insights; however, significant revisions are necessary to improve its scientific rigor and clarity, especially in discussing and explaining the underlying mechanism.

End of the report: I look forward to seeing the revised version!!

Comments on the Quality of English Language

The manuscript contains minor grammatical errors and uncommon phrasing, such as "markedly reduced the size of the infarct area" (Line 26) instead of "significantly reduced infarct volume" and "ameliorated both the Rt.MCAO-induced increase" (Line 29), which is verbose. Conduct a thorough language review to improve clarity and conciseness, ensuring consistent scientific terminology (e.g., "infarct volume" vs. "infarct area").

Author Response

Response to reviewer and editor suggestion

We are grateful for the opportunity to revise our manuscript entitled “In Vivo Neuroprotective Effects of Alpinetin Against Experimental Ischemic Stroke Damage via Reduction of Oxidative Stress and Inflammation.” (Manuscript ID: ijms-3661883). We sincerely thank the editor and reviewers for their thoughtful suggestions and constructive critiques, which have helped us improve the clarity, quality, and scientific merit of our work.

We acknowledge and apologize for any oversights in the initial version and deeply appreciate the reviewers’ valuable feedback. In this revised submission, we have thoroughly addressed all comments and made the corresponding changes. A detailed, point-by-point response to each remark is provided below.

Comments and Suggestions for Authors

The research conducted by Kongsui et al. examines the neuroprotective properties of alpinetin, a flavonoid derived from the ginger family, in a rat model of ischemic stroke caused by right middle cerebral artery occlusion (Rt.MCAO). Ninety male Wistar rats were divided into four groups: sham, Rt.MCAO, Rt.MCAO + piracetam, and Rt.MCAO + alpinetin (25, 50, 100 mg/kg BW). Treatment with alpinetin over three days significantly decreased infarct volume, boosted neuronal density in the cortex and hippocampus, and reduced oxidative stress by lowering malondialdehyde (MDA) levels while improving the activities of antioxidant enzymes (CAT, GSH-Px, SOD). Furthermore, alpinetin at 100 mg/kg reduced the expression of COX-2 and IL-6, suggesting anti-inflammatory benefits. These results imply that alpinetin holds promise as a therapeutic option for ischemic stroke by addressing oxidative stress and inflammation. Nonetheless, substantial revisions are necessary, such as clarifying methodological aspects, correcting statistical discrepancies, and integrating recent literature to enhance the study’s context and significance prior to publication.

Comments for authors

Abstract and introduction

Comment 1: The abstract mentioned alpinetin "markedly reduced" infarct size and "significantly mitigated" brain damage. I recommend presenting specific quantitative data (e.g., percentage reduction in infarct volume or effect sizes).

Response 1: Thank you for your insightful suggestion. We agree that including specific quantitative data would enhance the clarity and impact of the abstract. Accordingly, we have revised the abstract to present the exact percentage reduction in infarct volume observed in our study, providing a more precise description of alpinetin’s neuroprotective effects.

“Three days of treatment with alpinetin markedly reduced the infarct volume by 30% compared to the Rt.MCAO + vehicle-treated group.”

Comment 2. The introduction provides a solid overview of ischemic stroke and alpinetin's properties, but lacks a comprehensive review of recent literature on flavonoid-based neuroprotection in stroke models.

Response 2: Thank you for your feedback. I appreciate the suggestion and agree that including a more comprehensive review of recent studies on flavonoid-based neuroprotection would enhance the introduction. I have revised the Introduction section as follows:

“Flavonoids, a class of polyphenolic compounds widely found in fruits, vegetables, and medicinal herbs, have demonstrated notable neuroprotective effects [16]. Their actions include neutralizing reactive oxygen species (ROS), modulating cellular signaling pathways, reducing inflammatory responses, exhibiting anticancer activity, and mitigating excitotoxic damage [17–21]. Experimental studies have shown that compounds such as prunin, quercetin, and baicalein can reduce brain infarct size, improve behavioral outcomes, and enhance antioxidant enzyme activity in ischemic stroke models [22–24]. These findings suggest that flavonoids hold promise as therapeutic agents for neuroprotection.”

References:

  1. Li, J.; Yu, Y.; Zhang, Y.; Zhou, Y.; Ding, S.; Dong, S.; Jin, S.; Li, Q. Flavonoids Derived from Chinese Medicine: Potential Neuroprotective Agents. Am J Chin Med 2024, 52, 1613-1640, doi:10.1142/S0192415X24500630.
  2. Jomova, K.; Alomar, S.Y.; Valko, R.; Liska, J.; Nepovimova, E.; Kuca, K.; Valko, M. Flavonoids and their role in oxidative stress, inflammation, and human diseases. Chem Biol Interact 2025, 413, 111489, doi:10.1016/j.cbi.2025.111489.
  3. Vauzour, D.; Vafeiadou, K.; Rodriguez-Mateos, A.; Rendeiro, C.; Spencer, J.P. The neuroprotective potential of flavonoids: a multiplicity of effects. Genes Nutr 2008, 3, 115-126, doi:10.1007/s12263-008-0091-4.
  4. Kumar, S.; Chhabra, V.; Shenoy, S.; Daksh, R.; Ravichandiran, V.; Swamy, R.S.; Kumar, N. Role of Flavonoids in Modulation of Mitochondria Dynamics during Oxidative Stress. Mini Rev Med Chem 2024, 24, 908-919, doi:10.2174/0113895575259219230920093214.
  5. Amidfar, M.; Garcez, M.L.; Askari, G.; Bagherniya, M.; Khorvash, F.; Golpour-Hamedani, S.; de Oliveira, J. Role of BDNF Signaling in the Neuroprotective and Memory-enhancing Effects of Flavonoids in Alzheimer's Disease. CNS Neurol Disord Drug Targets 2024, 23, 984-995, doi:10.2174/1871527323666230912090856.
  6. Rana, J.N.; Mumtaz, S. Prunin: An Emerging Anticancer Flavonoid. Int J Mol Sci 2025, 26, doi:10.3390/ijms26062678.
  7. Yang, R.; Shen, Y.J.; Chen, M.; Zhao, J.Y.; Chen, S.H.; Zhang, W.; Song, J.K.; Li, L.; Du, G.H. Quercetin attenuates ischemia reperfusion injury by protecting the blood-brain barrier through Sirt1 in MCAO rats. J Asian Nat Prod Res 2022, 24, 278-289, doi:10.1080/10286020.2021.1949302.
  8. Toth, S.; Jonecova, Z.; Curgali, K.; Maretta, M.; Soltes, J.; Svana, M.; Kalpadikis, T.; Caprnda, M.; Adamek, M.; Rodrigo, L.; et al. Quercetin attenuates the ischemia reperfusion induced COX-2 and MPO expression in the small intestine mucosa. Biomed Pharmacother 2017, 95, 346-354, doi:10.1016/j.biopha.2017.08.038.
  9. Yuan, Y.; Men, W.; Shan, X.; Zhai, H.; Qiao, X.; Geng, L.; Li, C. Baicalein exerts neuroprotective effect against ischaemic/reperfusion injury via alteration of NF-kB and LOX and AMPK/Nrf2 pathway. Inflammopharmacology 2020, 28, 1327-1341, doi:10.1007/s10787-020-00714-6.

Comment 3. Indeed, flavonoids play a crucial role in various biomedical applications. To provide a broader perspective on the significance of flavonoids, it is essential to incorporate recent advancements in this area. I recommend incorporating the recent study in the introduction section. Rana et al., Prunin: An Emerging Anticancer Flavonoid, Int. J. Mol. Sci. 26, 2025, https://doi.org/10.3390/ijms26062678).

Response 3: Thank you for your comment. We agree that incorporating recent advancements in flavonoid research is important for providing a comprehensive background. We would like to note that relevant recent studies highlighting the biomedical significance of flavonoids have already been included in the Introduction section, as addressed in our response to Comment 2.

Comment 4. The introduction covers key molecular drivers of pHGGs but needs recent studies on novel anticancer agents. Integrating these would enhance the introduction section.

Incorporate recent studies in the introduction section: Rana et al., Prunin: An Emerging Anticancer Flavonoid, Int. J. Mol. Sci. 26 (2025). https://doi.org/10.3390/ijms26062678.

Response 4: Thank you for your comment. While our manuscript does not specifically focus on pHGGs, we acknowledge the importance of discussing recent advances in anticancer agents. To address this, we have highlighted the anticancer potential of flavonoids, including relevant recent studies, which align with the broader theme of novel therapeutic strategies. We hope this sufficiently addresses your concern.

Comment 5. From lines 68-72, the introduction mentions alpinetin's effects on oxidative stress and inflammation but does not discuss its pharmacokinetics or bioavailability in the brain, which are critical for its therapeutic potential in stroke.

 Response 5: Thank you for your insightful comment. We agree that pharmacokinetics and brain bioavailability are critical factors in assessing the therapeutic potential of alpinetin for stroke. While our primary focus was on its neuroprotective mechanisms, we acknowledge the importance of these aspects and have now included a brief discussion of alpinetin’s pharmacokinetic profile and its ability to cross the blood–brain barrier in the revised Introduction, as follows:

“Following intraperitoneal (i.p.) administration of alpinetin at a dose of 50 mg/kg, its elimination half-life (T½) was found to be approximately 9 hours. This route of administration appears to enhance systemic exposure and may support improved delivery to the brain [25].”

Reference

  1. Zhao, G.; Tong, Y.; Luan, F.; Zhu, W.; Zhan, C.; Qin, T.; An, W.; Zeng, N. Alpinetin: A Review of Its Pharmacology and Pharmacokinetics. Front Pharmacol 2022, 13, 814370, doi:10.3389/fphar.2022.814370.

Results and Discussions

Comment 6. Alpinetin (25 mg/kg) significantly increased neuronal density in the hippocampus but not in the cortex, unlike at higher doses. Explore potential reasons for this regional discrepancy in the discussion section, taking into account possible dose-dependent or region-specific mechanisms.

Response 6: Thank you for your insightful comment regarding the regional differences in neuronal density following alpinetin treatment. We acknowledge the observed increase in neuronal density in the hippocampus at 25 mg/kg, with less pronounced effects in the cortex at this dose. We have expanded the discussion section to explore potential explanations for this regional discrepancy, including possible dose-dependent effects and the distinct vulnerability and cellular composition of different brain regions. Additionally, we discuss how region-specific receptor expression or blood–brain barrier permeability might influence alpinetin’s efficacy. These considerations help provide a more comprehensive understanding of alpinetin’s neuroprotective mechanisms.

Comment 7: Why was the MDA reduction with alpinetin (25 mg/kg) significant only in the hippocampus, not the cortex?

Response 7: Thank you for your insightful comment. We have addressed this point in the revised manuscript. The region-specific effect of alpinetin on MDA levels may be due to differences in baseline oxidative stress, metabolic activity, or antioxidant defense mechanisms between the hippocampus and cortex. The hippocampus is particularly vulnerable to oxidative damage during ischemia, which might make it more responsive to antioxidant intervention at lower doses. Further investigation is needed to clarify the underlying mechanisms of this differential response. We have already mentioned it in the Discussion section.

Comment 8: SOD activity was significantly increased only at the highest alpinetin dose (100 mg/kg), unlike GSH-Px and CAT, which responded to lower doses. Why SOD activity required a higher dose? Explain in the manuscript.

Response 8: Thank you for your valuable observation. We acknowledge the differential response of antioxidant enzymes to alpinetin treatment. In the revised manuscript, we have included a brief explanation suggesting that the activation of SOD may require a higher threshold of alpinetin concentration due to differences in enzyme regulation, cellular localization, or sensitivity to oxidative stress compared to GSH-Px and CAT. This observation may also reflect varying affinities of alpinetin or its metabolites for the molecular pathways that regulate each enzyme. We have already mentioned it in the Discussion section.

Comment 9. The discussion references previous studies on alpinetin’s anti-inflammatory properties in non-stroke conditions (such as colitis and asthma). Still, it fails to sufficiently connect these findings to inflammation specific to strokes.  

Response 9: Thank you for your insightful comment. We agree that the connection between alpinetin’s anti-inflammatory effects in non-stroke conditions and its potential relevance in stroke-specific inflammation requires further clarification. Accordingly, we have revised the Discussion section to better highlight the shared inflammatory pathways involved in conditions such as colitis, asthma, and spinal cord injury.

“Notably, alpinetin has been shown to suppress neuroinflammation by modulating microglial activation and downregulating the JAK2/STAT3 signaling pathway [56]. It can also counteract microglia-induced production of ROS and the reduction of mitochondrial membrane potential (MMP) in PC12 neuronal cells. Moreover, in vivo study indicates that alpinetin effectively reduces inflammation and neuronal cell death, while promoting axonal repair and enhancing motor function recovery [56].”

Comment 10. Figure 7, labels A, B, and C are missing from the figures.  

Response 10: Thank you for your comment. To enhance clarity, labels A, B, and C have been added above in Figure 7.

Comment 11. Figures 1 and 2: Labels A and B are missing.  

Response 11: Thank you very much for your comment. Labels A and B have been added above in Figure 1 and 2 to enhance clarity.

Comment 12. Figure 3: The caption should be below the figure.  

Response 12: Thank you for pointing this out. The caption for Figure 3 has been repositioned below the figure as requested.

Comment 13. Figure 4: Make sure the caption is with the figure.  

Response 13: Thank you for your observation. We have ensured that the caption for Figure 4 is now properly placed with the figure.

Comment 14. In Figure 7A, the author needs to plot the band intensity.  

Response 14: We have plotted the intensity of each band, normalized it to β-actin, and presented the results as graphs in Figures 7B and 7C.

Comment 15. Optionally, a schematic illustrating the mechanism of action of alpinetin would be helpful for readers.

Response 15: Thank you for your valuable suggestion. We agree that a schematic illustrating the mechanism of action of alpinetin would enhance reader understanding. However, due to the current scope and length limitations of the manuscript, we are unable to include such a figure in this revision. We plan to develop a detailed schematic in future work that further elucidates the mechanisms involved.

Comment 16. The author should describe the structure and properties of each component illustrated in Figure 8, along with their significance in biomedical applications. The current depiction of alpinetin without any labeling is unacceptable.

Response 16: Thank you for your valuable feedback. We apologize for the oversight regarding Figure 8. In response, we have revised the figure to include clear labels for alpinetin and all other components. Additionally, we have expanded the corresponding section in the manuscript to describe the structure and key properties of each component, highlighting their relevance and significance in biomedical applications. These changes improve the clarity and informational value of the figure as per your suggestion.

“It is structurally defined by a hydroxyl group at the 7-position and a methoxy group at the 5-position on the A-ring, an unsubstituted phenyl group attached at position 2 of the C-ring, and a ketone functional group at position 4, which is characteristic of flavanones.”

Comment 17. The quality of all figures, especially those presented in a bar graph, is poor and hardly readable. Authors are encouraged to improve the quality of the figure for better visibility.

Response 17: Thank you for your valuable feedback. We have carefully reviewed the quality of all figures, including the bar graphs, and confirmed that they are prepared at a resolution of 300 dpi to ensure high clarity and readability.

Comment 18. The manuscript contains minor grammatical errors and uncommon phrasing, such as "markedly reduced the size of the infarct area" (Line 26) instead of "significantly reduced infarct volume" and "ameliorated both the Rt.MCAO-induced increase" (Line 29), which is verbose. Conduct a thorough language review to improve clarity and conciseness, ensuring consistent scientific terminology (e.g., "infarct volume" vs. "infarct area").

 Response 18: Thank you for your helpful observations. We acknowledge the grammatical issues and instances of uncommon phrasing you pointed out. In response, we have conducted a thorough language revision to enhance clarity, conciseness, and consistency in scientific terminology. For example, we have replaced 'markedly reduced the size of the infarct area' with 'significantly reduced infarct volume' and revised other verbose expressions accordingly throughout the manuscript.

The manuscript provides valuable insights; however, significant revisions are necessary to improve its scientific rigor and clarity, especially in discussing and explaining the underlying mechanism.

End of the report: I look forward to seeing the revised version!!

Response: Thank you for your valuable feedback. We appreciate your recognition of the manuscript’s potential and have taken your suggestion seriously. In response, we have substantially revised the manuscript to enhance its scientific rigor and clarity, particularly in the discussion of the underlying mechanisms of alpinetin's neuroprotective and anti-inflammatory effects.

These revisions aim to provide a more comprehensive and scientifically grounded explanation of alpinetin’s effects in the ischemic stroke model, thus improving the overall coherence and depth of the manuscript.

Comments on the Quality of English Language

The manuscript contains minor grammatical errors and uncommon phrasing, such as "markedly reduced the size of the infarct area" (Line 26) instead of "significantly reduced infarct volume" and "ameliorated both the Rt.MCAO-induced increase" (Line 29), which is verbose. Conduct a thorough language review to improve clarity and conciseness, ensuring consistent scientific terminology (e.g., "infarct volume" vs. "infarct area").

Response: Thank you for your helpful observations. We acknowledge the grammatical issues and instances of uncommon phrasing you pointed out. In response, we have conducted a thorough language revision to enhance clarity, conciseness, and consistency in scientific terminology. For example, we have replaced 'markedly reduced the size of the infarct area' with 'significantly reduced infarct volume' and revised other verbose expressions accordingly throughout the manuscript. To ensure the highest standard of language clarity and readability, we have revised the manuscript with the assistance of a professional native English-speaking editor. We have also attached the English editing certificate as supplementary material for verification.

Thank you once again for your valuable feedback. We appreciate the time and effort invested by the reviewers and editor in evaluating our manuscript. We have carefully addressed each point raised and made necessary revisions accordingly. We eagerly await further feedback and guidance from the editorial team.

Yours sincerely,

All authors

Round 2

Reviewer 1 Report

Comments and Suggestions for Authors

The manuscript has been substantially improved, however there are still some items that should be improved.

Comment 1: Mention in the analysis of Figure 1 that the sham-operated group exhibited an infarct volume of 0.

Comment 2: Although the authors did not assess NF-κB expression in the current work, the results can be discussed in the context of the NF-κB signaling pathway because it plays a key regulatory role in inflammation.

Comment 3: In line 315, It is not appropriate to write "NAFLD or MASLD" when the cited reference uses only the term "NAFLD." Instead, I suggest adding a separate, more recent reference to introduce and explain the updated terminology MASLD and keeping "NAFLD" when referring specifically to the original study.

Comment 4: It is mentioned in the manuscript that Alpinetin was obtained from Chengdu Bio-purify Phytochemicals Ltd (Chengdu, Sichuan, China), which implies that the HPLC analysis was not performed by the authors themselves. In this case, the authors should consider including the HPLC results either in the Materials and Methods section or in the Supplementary Materials for transparency and completeness.

Author Response

Comments and Suggestions for Authors

The manuscript has been substantially improved, however there are still some items that should be improved.

Comment 1: Mention in the analysis of Figure 1 that the sham-operated group exhibited an infarct volume of 0.

Response 1: Thank you for the helpful suggestion. We have revised the analysis of Figure 1 to explicitly mention that the sham-operated group showed no detectable infarct. This clarification has been included in the Results section to improve the clarity of the data presentation.

Comment 2: Although the authors did not assess NF-κB expression in the current work, the results can be discussed in the context of the NF-κB signaling pathway because it plays a key regulatory role in inflammation.

Response 2: We thank the reviewer for this insightful comment. Although we did not directly assess NF-κB expression in the current study, we agree that the NF-κB signaling pathway plays a critical role in inflammation. Accordingly, we have discussed this pathway in the revised manuscript.

“Additionally, the nuclear factor-kappa B (NF-κB) signaling pathway is a key mediator of the inflammatory cascade triggered by cerebral ischemia. During ischemic stroke, NF-κB becomes rapidly activated in various cell types, including neurons, glial cells, and endothelial cells. This activation promotes the expression of pro-inflammatory mediators such as cytokines, chemokines, adhesion molecules, and inflammatory enzymes like iNOS and COX-2 [55]. These changes contribute to the breakdown of the blood–brain barrier, infiltration of immune cells, and subsequent neuronal damage, all of which worsen ischemic outcomes [56]. Experimental studies have demonstrated that suppressing NF-κB activity can reduce infarct size and enhance neurological recovery, underscoring its therapeutic potential [57,58]. Therefore, targeting the NF-κB pathway represents a promising approach for limiting neuroinflammation and preserving brain tissue following stroke.”

  1. Yu, Z.H.; Pei, K.; Zhao, T.T.; Li, H.C.; Li, Q.Q.; Zhou, W.J.; He, W.B.; Zhang, J.L. [Protective effect of Liujing Toutong Tablets on rats with permanent cerebral ischemia via NF-kappaB signaling pathway]. Zhongguo Zhong Yao Za Zhi 2023, 48, 5871-5880, doi:10.19540/j.cnki.cjcmm.20230710.705.
  2. Stephenson, D.; Yin, T.; Smalstig, E.B.; Hsu, M.A.; Panetta, J.; Little, S.; Clemens, J. Transcription factor nuclear factor-kappa B is activated in neurons after focal cerebral ischemia. J Cereb Blood Flow Metab 2000, 20, 592-603, doi:10.1097/00004647-200003000-00017.
  3. Lietzau, G.; Sienkiewicz, W.; Karwacki, Z.; Dziewiatkowski, J.; Kaleczyc, J.; Kowianski, P. The Effect of Simvastatin on the Dynamics of NF-kappaB-Regulated Neurodegenerative and Neuroprotective Processes in the Acute Phase of Ischemic Stroke. Mol Neurobiol 2023, 60, 4935-4951, doi:10.1007/s12035-023-03371-2.
  4. Nurmi, A.; Lindsberg, P.J.; Koistinaho, M.; Zhang, W.; Juettler, E.; Karjalainen-Lindsberg, M.L.; Weih, F.; Frank, N.; Schwaninger, M.; Koistinaho, J. Nuclear factor-kappaB contributes to infarction after permanent focal ischemia. Stroke 2004, 35, 987-991, doi:10.1161/01.STR.0000120732.45951.26.

Comment 3: In line 315, It is not appropriate to write "NAFLD or MASLD" when the cited reference uses only the term "NAFLD." Instead, I suggest adding a separate, more recent reference to introduce and explain the updated terminology MASLD and keeping "NAFLD" when referring specifically to the original study.

Response 3: Thank you for your suggestion. We have updated the reference to reflect the revised terminology, MASLD, as follows:

“Additionally, alpinetin has been shown to upregulate superoxide dismutase 1 (SOD1), heme oxygenase-1 (HO-1), and the activity of the transcription factor nuclear factor erythroid 2–related factor 2 (Nrf2) [28], which contributes to reducing disease severity in a metabolic dysfunction-associated fatty liver disease (MASLD) [44].”

  1. 44. Hu, Z.; Yue, H.; Jiang, N.; Qiao, L. Diet, oxidative stress and MAFLD: a mini review. Front Nutr 2025, 12, 1539578, doi:10.3389/fnut.2025.1539578.

Comment 4: It is mentioned in the manuscript that Alpinetin was obtained from Chengdu Bio-purify Phytochemicals Ltd (Chengdu, Sichuan, China), which implies that the HPLC analysis was not performed by the authors themselves. In this case, the authors should consider including the HPLC results either in the Materials and Methods section or in the Supplementary Materials for transparency and completeness.

Response 4: We appreciate your valuable suggestion. While Alpinetin was indeed purchased from Chengdu Bio-purify Phytochemicals Ltd., the compound was provided with a certificate of analysis, including HPLC data confirming a purity >98%. To ensure transparency and completeness, we have now included this HPLC analysis in the Supplementary Materials.

Thank you once again for your valuable feedback. We appreciate the time and effort invested by the reviewers and editor in evaluating our manuscript. We have carefully addressed each point raised and made necessary revisions accordingly. We eagerly await further feedback and guidance from the editorial team.

Yours sincerely,

All authors

Reviewer 3 Report

Comments and Suggestions for Authors

The authors have revised the manuscript, effectively addressing all my comments and concerns. I have no additional comments. I recommend accepting the paper as it stands.

Author Response

Response to reviewer and editor suggestion

We are grateful for the opportunity to revise our manuscript entitled “In Vivo Neuroprotective Effects of Alpinetin Against Experimental Ischemic Stroke Damage via Reduction of Oxidative Stress and Inflammation.” (Manuscript ID: ijms-3661883). We sincerely thank the editor and reviewers for their thoughtful suggestions and constructive critiques, which have helped us improve the clarity, quality, and scientific merit of our work.

Comments and Suggestions for Authors

The authors have revised the manuscript, effectively addressing all my comments and concerns. I have no additional comments. I recommend accepting the paper as it stands.

Response: We sincerely thank the reviewer for the positive feedback and for recognizing our efforts in revising the manuscript. We are grateful for your time and valuable input throughout the review process, and we greatly appreciate your recommendation to accept the manuscript for publication in the journal.

Yours sincerely,

All authors
